# POINT-BIND & POINT-LLM: ALIGNING POINT CLOUD WITH MULTI-MODALITY FOR 3D UNDERSTANDING, GENERATION, AND INSTRUCTION FOLLOWING

## ABSTRACT

With the growing diversity of large-scale data, learning from multi-modality has attained notable progress in language and 2D vision. However, in 3D domains, how to develop an all-purpose multi-modal framework is still under-explored. To this end, we introduce **Point-Bind**, a 3D multi-modality model aligning point clouds with 2D image, language, audio, and video. Guided by ImageBind, we construct a joint embedding space between 3D and multi-modalities, enabling many promising applications, e.g., 3D embedding arithmetic, any-to-3D generation, and 3D open-world understanding. On top of this joint embedding space, we further present **Point-LLM**, a 3D large language model extending ImageBind-LLM to follow 3D and multi-modal instructions. *Without any 3D instruction data*, our Point-LLM injects the semantics of Point-Bind into pre-trained LLMs, e.g., LLaMA, and exhibits superior 3D and multi-modal question-answering capacity. We have conducted extensive experiments to demonstrate the effectiveness and generalizability of our approach for aligning 3D and multi-modality.

## 1 INTRODUCTION

In these years, 3D vision has gained significant attention and development, driven by the rising popularity of autonomous driving (Chen et al., 2020b; Shi et al., 2020), navigation (Tan et al., 2001; Wang et al., 2019), 3D scene understanding (Armeni et al., 2016; Liu et al., 2021b), and robotics (Huang et al., 2023; Savva et al., 2019). To extend its application scenarios, numerous efforts have been made to incorporate 3D point clouds with other modalities, allowing for improved 3D understanding (Guo et al., 2023a; Afham et al., 2022), text-to-3D generation (Nichol et al., 2022; Poole et al., 2022), and 3D question answering (Azuma et al., 2022; Hong et al., 2023a).

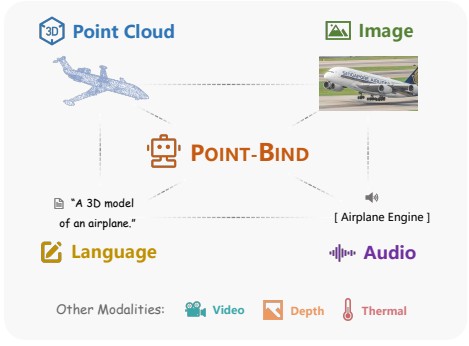

Figure 1: **Overview of Point-Bind.** We propose a unified and general framework to align 3D with multiple modalities.

For 3D geometry understanding, previous works either leverage 2D-language embeddings to guide 3D open-world recognition (Zhang et al., 2022b), or harness visual and textual semantics to assist 3D representation learning (Xue et al., 2022). However, their perception capabilities are mostly constrained by limited modalities provided in the training phase. Inspired by 2D generative models, a collection of methods (Lin et al., 2023; Nichol et al., 2022) has achieved text-to-3D synthesis with high quality and efficiency. Despite this, they lack the ability to generate 3D shapes conditioned on multi-modal input, e.g., a sound and an image. Another series of works connects descriptive natural language with 3D data, applying to 3D captioning (Yuan et al., 2022; Chen et al., 2023b) and question answering (Wijmans et al., 2019; Azuma et al., 2022). Yet, they fail to utilize the pre-trained linguistic knowledge within large language models (LLMs) to better reason 3D geometries. Therefore, how to develop a unified 3D framework aligning with multi-modality for general 3D learning still remains an open question. Very recently, ImageBind (Girdhar et al., 2023) is proposed to learn a shared representation space across six different modalities, i.e., image, text, audio,

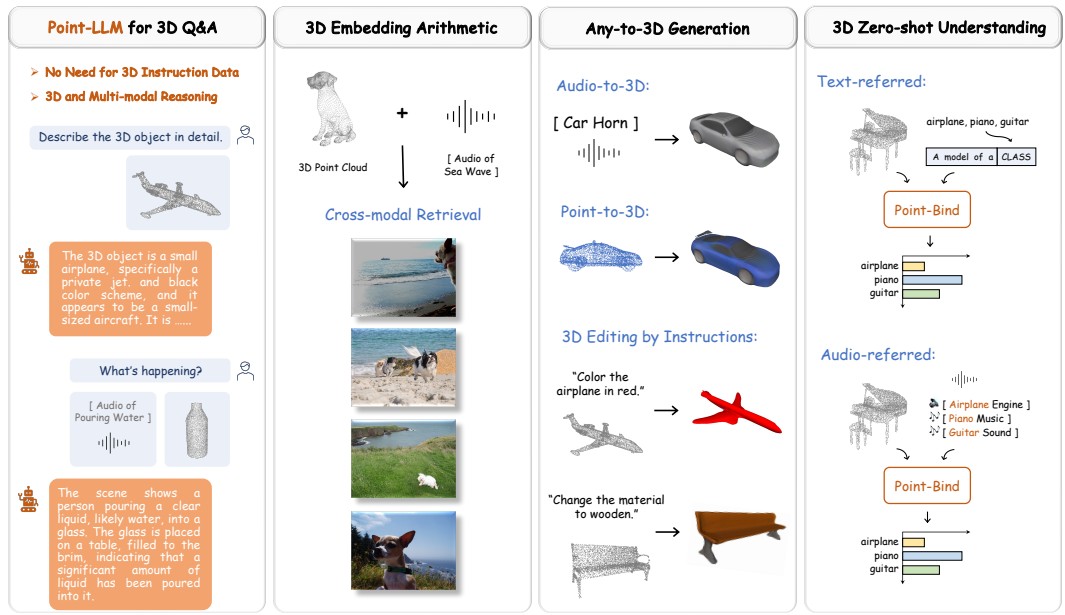

Figure 2: **3D Multi-modal Applications of Point-Bind.** With a joint 3D multi-modal embedding space, Point-Bind enables many promising application scenarios, e.g., Point-LLM for 3D instruction following, 3D generation conditioned on any modalities, embedding-space arithmetic with 3D, and multi-modal 3D zero-shot understanding.

depth, thermal, and IMU data. Motivated by this, we ask the following question: *can we construct a joint embedding space between 3D and multi-modality for unified 3D understanding, generation, and instruction following?*

In this paper, we introduce **Point-Bind**, a 3D multi-modality framework that aligns point cloud with multiple modalities for general 3D analysis, as shown in Figure 1. Specifically, we first collect 3D-image-text-audio pairs as the training data, and learn a joint embedding space guided by Image-Bind, or other multi-modal large models (Zhu et al., 2023a). Based on the pre-training paragdim of previous works (Xue et al., 2022; Zeng et al., 2023), we adopt a contrastive loss between the extracted features from a trainable 3D encoder, e.g., I2P-MAE (Zhang et al., 2023a), and the pre-trained multi-modal encoders. In this way, we efficiently integrate different modalities into a unified representation space, which also includes modalities that are absent during training, such as video, depth, and infrared data. The joint space of Point-Bind is expected to expand the scope of 3D models to wider cross-modal scenarios.

On top of this, Point-Bind naturally motivates several emergent 3D-centric multi-modal applications, as shown in Figure 2. Note that, such *emergent* characteristics can alleviate the need for expensive task-specific training, significantly lowering the bar to efficiently achieve new 3D cross-modal tasks, summarized as follows:

- **3D Embedding-space Arithmetic.** We observe the encoded 3D features from Point-Bind can be added with other modalities to incorporate their semantics, achieving favorable composed cross-modal retrieval performance.

- **Any-to-3D Generation.** Based on existing text-to-3D generative models, Point-Bind enables 3D shape synthesis conditioned on any input modalities and their composition, e.g., text/image/audio/point-to-mesh, or editing 3D shapes with multi-modal instructions.

- **3D Open-world Understanding.** Benefiting from multi-modal semantics, Point-Bind attains leading performance for 3D zero-shot classification, referred to by text. Also, our approach supports audio-referred 3D open-world understanding with satisfactory results.

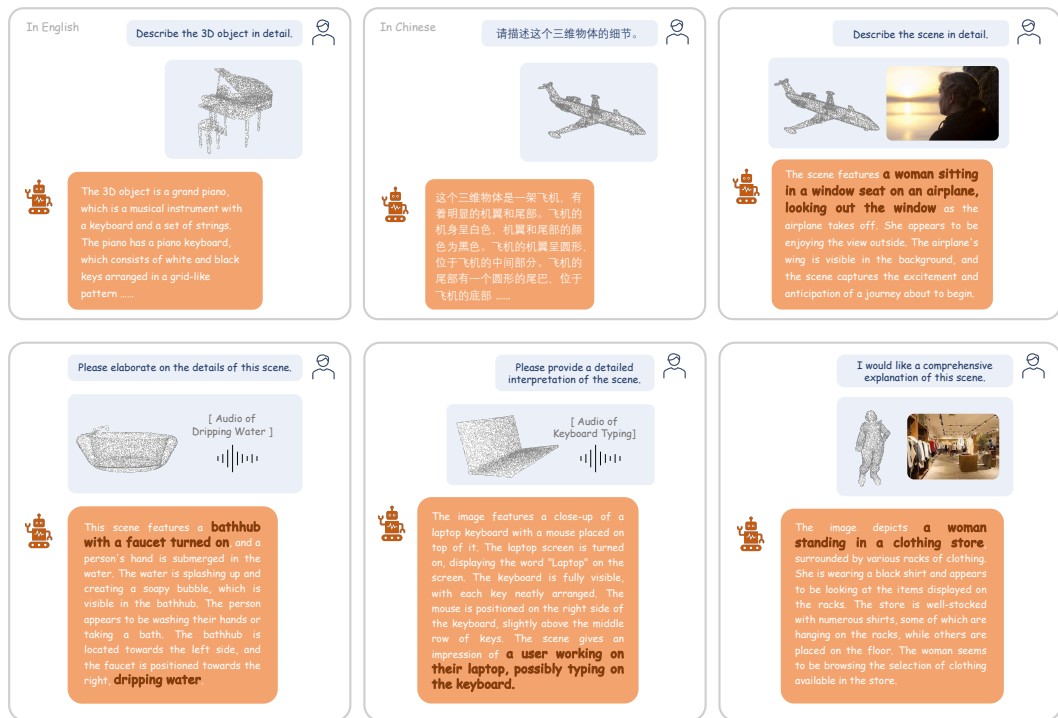

Figure 3: **3D Question-answering Examples of Point-LLM.** Given 3D and multi-modal instructions, our Point-LLM can effectively generate detailed responses and conduct superior cross-modal reasoning. Notably, we do not need any 3D instruction data for training.

Furthermore, with the joint embedding space, we propose to incorporate Point-Bind with the pre-trained ImageBind-LLM (Han et al., 2023) to develop a 3D large language model, termed as **Point-LLM**. As shown in Figure 3, our Point-LLM can respond to language instructions with 3D point cloud conditions, and effectively capture spatial geometry characteristics with bilingual competence. Referring to ImageBind-LLM, we connect our Point-Bind with its pre-trained bind network and visual cache model to bridge our 3D embedding space with LLaMA (Touvron et al., 2023). In such a training-free manner, our Point-LLM enables LLaMA to understand the 3D world with superior question-answering capacity, while *requiring no 3D instruction data*. Notably, our approach can generate descriptive responses conditioned on a combination of 3D and multi-modal input, e.g., a point cloud with an image/audio, indicating strong cross-modal reasoning capacity.

## 2 POINT-BIND

The overall pipeline of Point-Bind is shown in Figure 4. In Section 2.1, we first provide a preliminary of ImageBind (Girdhar et al., 2023). Then, in Section 2.2 and 2.3, we elaborate on the training data and multi-modal alignment for Point-Bind, respectively. Finally, in Section 2.4, we introduce several 3D-centric applications derived from our approach.

### 2.1 PRELIMINARY OF IMAGEBIND

ImageBind proposes an approach to combine multiple modalities together, which utilizes only image-paired data to learn a joint embedding space of six modalities, i.e., images, text, audio, depth, thermal, and IMU data. It does not need training dataset pairing all six modalities, but leverages the binding property of 2D images, i.e., aligning every single modality to image independently. Specifically, ImageBind feeds multi-modal input into corresponding encoders, and adopts for cross-modal contrastive learning. After training on large-scale image-paired data, ImageBind effectively aligns six modalities into a single representation space, enabling emergent cross-modal capabilities.

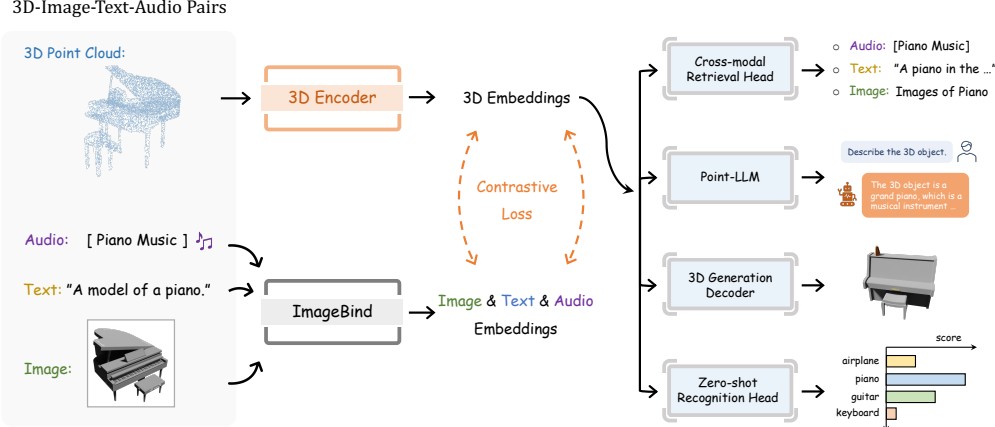

Figure 4: **Overall Pipeline of Point-Bind.** We collect 3D-image-audio-text data pairs for contrastive learning, which aligns 3D with other modalities guided ImageBind (Girdhar et al., 2023). With a joint embedding space, Point-Bind can be utilized for 3D cross-modal retrieval, any-to-3D generation, 3D zero-shot understanding, and developing a 3D large language model, Point-LLM.

Inspired by this, we propose to develop a 3D multi-modal framework, Point-Bind, which leverages ImageBind, or its follow-up work (Zhu et al., 2023a), as guidance to incorporate 3D point cloud with other modalities for general 3D understanding, generation, and instruction following.

## 2.2 TRAINING DATA

To align 3D with multi-modalities, we leverage the pre-trained joint embedding space of Image-Bind (Girdhar et al., 2023) and adopt contrastive loss (Zhang et al., 2022c; Radford et al., 2021) to simultaneously align 3D point clouds with the other three modalities: image, text, and audio. To obtain the contrastive training data, we collect a cross-modal dataset of 3D-image-audio-text pairs. There are three steps for dataset collection as follows.

**3D-image-text Pairs.** We adopt the data pairs of 3D, images, and text from ULIP (Xue et al., 2022), which includes 3D-image-text triplets built from ShapeNet (Chang et al., 2015), a common-used dataset containing abundant 3D CAD models. Each 3D point cloud is paired with a corresponding text describing the semantic information of its spatial shape, and a 2D counterpart generated by multi-view image rendering. The text description is constructed by a synset of category names and 64 pre-defined templates.

**3D-audio Pairs.** To provide more contrastive signals from a fourth modality, we collect the data pairs of 3D and audio from ESC-50 (Piczak, 2015) and ShapeNet datasets. Specifically, we first select the categories whose objects can make a sound in the real world from the 55 categories of ShapeNet, such as 'airplane', 'clock', 'washing machine', and 'keyboard'. Then, we preserve only the categories that are also within ESC-50. By this standard, we obtain 9 categories of 3D point clouds paired with extensive audio clips, i.e., 'airplane', 'chirping birds', 'can opening', 'car horn', 'clock tick', 'keyboard typing', 'crackling fire', and 'train'. Each category contains 40 audio samples, with a total number of 360. During training, for a point cloud within the nine categories, we randomly sample an audio sample and adopt data augmentation, e.g., random cropping and volume perturbation, for more robust training.

**3D-image-audio-text Pairs Construction.** Finally, we match each 3D-audio pair with its corresponding 3D-image-text data, resulting in a unified 3D-image-audio-text dataset with extensive cross-modal pairs. During training, we simultaneously feed point clouds and their paired data of three modalities for contrastive learning.

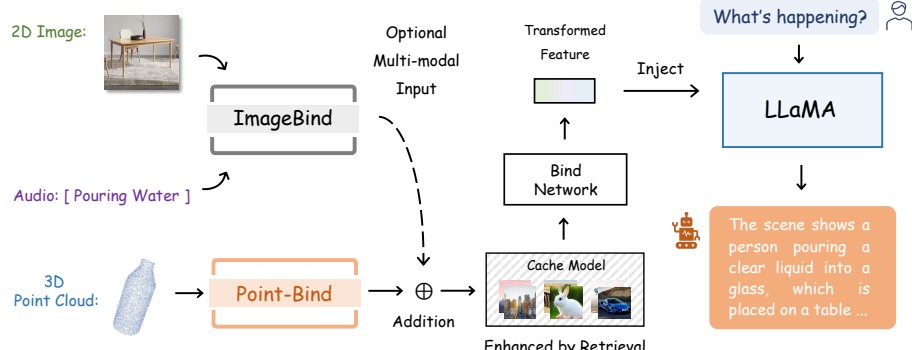

Figure 5: **Inference Paradigm of Point-LLM.** Due to our 3D joint embedding space, we can directly connect Point-Bind with a pre-trained bind network of ImageBind-LLM (Han et al., 2023) to enable LLaMA (Touvron et al., 2023) to follow 3D instructions. Optionally, our Point-LLM can also take as input multi-modality data, and conduct cross-modal reasoning for language response.

## 2.3 ALIGNING 3D WITH MULTI-MODALITY

After collecting the 3D paired data, we conduct contrastive training to learn a joint embedding space aligning 3D and multi-modalities. Each data sample contains a point cloud $P$, along with the paired 2D image $I$, text description $T^s$, and audio $A$, where $T^s$ represents a set of 64 pre-defined templates. For the point cloud, we adopt I2P-MAE (Zhang et al., 2023a) as the learnable 3D encoder, denoted as $\text{Encoder}_{3D}(\cdot)$, and append a projection network $\text{Proj}(\cdot)$ of two linear layers, which transforms the encoded 3D feature into ImageBind's multi-modal embedding space. We formulate it as

$$F_{3D} = \text{Proj}(\text{Encoder}_{3D}(P)), \tag{1}$$

where $F_{3D} \in \mathbb{R}^{1 \times C}$ denotes the projected 3D embedding, and $C$ equals the feature dimension of ImageBind. For the paired image-text-audio data, we leverage their corresponding encoders from ImageBind for feature extraction, which are frozen during training, formulated as

$$F_{2D}, F_T^s, F_A = \text{ImageBind}(I, T^s, A), \tag{2}$$

where $F_{2D}, F_A \in \mathbb{R}^{1 \times C}$ denote the image and audio embeddings, and $F_T^s \in \mathbb{R}^{64 \times C}$ denotes the text embedding for a set of 64 descriptions. Then, we conduct an average pooling as

$$F_T = \text{Average}(F_T^s) \quad \in \mathbb{R}^{1 \times C}, \tag{3}$$

which represents the aggregated text embedding with more robustness. After that, we adopt contrastive loss (Zhang et al., 2022c) between 3D and other modalities, which effectively enforces 3D embeddings to align with the joint representation space, formulated as

$$L_{total} = L(F_{3D}, F_{2D}) + L(F_{3D}, F_T) + L(F_{3D}, F_A).$$

Note that some training categories do not include the paired audio $A$, since they inherently cannot make any sound, e.g., bottle, planter, and couch, for which we ignore their audio features and loss.

## 2.4 MULTI-MODAL APPLICATIONS

Starting from the joint embedding space of Point-Bind, we introduce several emergent application scenarios concerning 3D and multi-modalities. Importantly, these new tasks are naturally ***emergent*** from Point-Bind, which means we do not need to spend many resources on task-specific training. Such characteristics significantly lower the bar for efficiently achieving new 3D cross-modal tasks.

**3D Embedding-space Arithmetic.** We observe that 3D features encoded by Point-Bind can be directly added with other modalities to incorporate their semantics, further achieving composed cross-modal retrieval. For instance, the combined embeddings of a 3D car and audio of sea waves can accurately retrieve an image showing a car parking by a beach, while the composition of a 3D laptop and audio of keyboard typing can retrieve an image of someone who is working with a laptop.

Figure 6: **Quantitative Evaluation of Point-LLM** evaluated by GPT-4 (Ope-nAI, 2023) and Bard (Google, 2023).

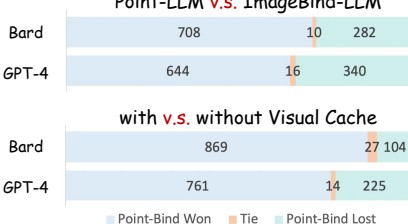

Table 1: **Performance on 3D Cross-modal Retrieval**, including 3D-to-3D, 2D-to-3D, 3D-to-2D, and text-to-3D retrieval. We report the mAP scores on the Model-Net40 dataset.

| Method | $3D \rightarrow 3D$ | $2D \rightarrow 3D$ | $3D \rightarrow 2D$ | $Text \rightarrow 3D$ |
|---|---|---|---|---|
| PointCLIP | 37.63 | 13.12 | 5.28 | 10.86 |
| PointCLIP-V2 | 47.94 | 20.48 | 9.22 | 52.73 |
| ULIP | 60.58 | 20.30 | 29.75 | 50.51 |
| ULIP-2 | **64.35** | 19.21 | 31.63 | 57.05 |
| **Point-Bind** | 63.23 | **34.59** | **42.83** | **64.50** |

**Any-to-3D Generation.** Existing 3D generation methods can only achieve text-to-3D synthesis. In contrast, with the joint embedding space of Point-Bind, we can generate the 3D mesh conditioned on any modalities, and also modify the appearance of existing 3D shapes with multi-modal instructions. In detail, we simply utilize a learnable projection layer to align our joint embedding space with the pre-trained decoder of existing 3D generation methods, e.g., ISS (Liu et al., 2022) by single view reconstruction (SVR). We only tune the projection layer while keeping other networks frozen. After this, we directly connect the multi-modal encoders of Point-Bind with the decoders of ISS, which is capable of synthesizing a 3D car mesh based on an input car horn. Also, we can feed an existing 3D shape using Point-Bind's 3D encoder, and provide multi-modal instruction signals to modify its appearance, e.g., coloring an airplane in red or changing a chair's material to wooden.

**3D Zero-shot Understanding.** For traditional text-inferred 3D zero-shot classification, Point-Bind attains *state-of-the-art* performance guided by additional multi-modal supervision. Besides, Point-Bind can also achieve audio-referred 3D open-world understanding, i.e., recognizing 3D shapes of novel categories indicated by the corresponding audio data (Piczak, 2015).

## 3 POINT-LLM

In this section, we illustrate how to leverage the emergent characteristic of Point-Bind to develop 3D large language models (LLMs), termed as Point-LLM, which enables a pre-trained ImageBind-LLM (Han et al., 2023) to achieve 3D question answering and multi-modal reasoning in a training-free manner. The overall pipeline of Point-LLM is shown in Figure 3.

**3D Instruction-following Capacity.** Our Point-LLM is developed on top of a pre-trained ImageBind-LLM (Han et al., 2023), which conducts multi-modality instruction tuning by injecting the semantics of ImageBind into LLaMA. By vision-language pre-training, ImageBind-LLM has already aligned the joint embedding space of ImageBind with LLaMA using a bind network, which shares the same space with our Point-Bind. Considering this, we can directly bridge Point-Bind with LLaMA using the pre-trained bind network. Therefore, our Point-LLM does not require any 3D instruction data for training, and efficiently endows LLaMA with 3D understanding capability, which also inherits the pre-trained bilingual ability of ImageBind-LLM. This significantly saves the resources for annotating 3D instruction data and large-scale training,

**Inference with Visual Cache.** For a given language instruction and a 3D point cloud, we input them into LLaMA and our Point-Bind, respectively. Then, before feeding the encoded 3D feature into the bind network, we adopt a visual cache model for 3D feature enhancement. As ImageBind-LLM adopts the image encoder of ImageBind for training, but we switch to Point-Bind's 3D encoder for inference, the cache model is designed to alleviate such 2D-3D modality discrepancy for better 3D geometry understanding. Specifically, the cache model stores three million ImageBind-encoded image features from the training data, which are regarded as both keys and values for knowledge retrieval. We regard the input 3D feature as the query, and retrieve the top-$k$ similar visual keys from the cache. Then, according to the cosine similarity, we aggregate the corresponding cached values (top-$k$ similar image features), and add the result to the original 3D feature via a residual connection. The enhanced 3D feature can adaptively incorporate similar 2D semantics from the cache model. Such a strategy mitigates the semantic gap of 2D-3D encoders, and boosts the representation qual-

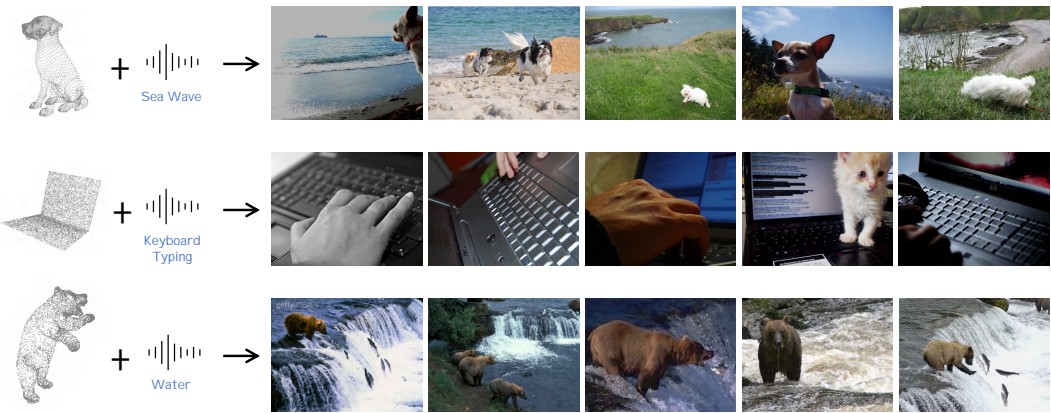

Figure 7: **Embedding-space Arithmetic of 3D and Audio.** To demonstrate our semantic composition ability, we retrieve 2D images with a combination of 3D point cloud and audio embeddings.

ity of 3D shapes in Point-LLM. After this, the enhanced feature is fed into the bind network for transformation and LLaMA for response generation.

**3D and Multi-modal Reasoning.** In addition to point clouds, considering the joint embedding space of Point-Bind, our Point-LLM can also conduct cross-modal reasoning and generate responses conditioned on multiple modalities. For an additional input image or audio, we utilize the image or audio encoder of ImageBind to extract the features, and directly add them with the 3D feature encoded by Point-Bind. By injecting such integrated features into LLaMA, Point-LLM can reason cross-modal semantics, and respond with the information of all input modalities. This demonstrates the promising significance of aligning multi-modality with 3D LLMs.

## 4 EXPERIMENTS

In this section, we respectively illustrate the emergent multi-modal applications of Point-Bind, i.e., Point-LLM for 3D instruction following, composed 3D cross-modal retrieval, any-to-3D generation, and 3D zero-shot understanding. Then, we conduct ablation studies to verify the effectiveness of our designs. Please refer to Supplementary Material for training details of Point-Bind.

### 4.1 POINT-LLM FOR 3D Q&A

**Settings.** Our method is built on a pre-trained ImageBind-LLM (Han et al., 2023), which adopts LLaMA 7B (Touvron et al., 2023) as the foundation LLM and ImageBind with a ViT-H image encoder. Note that we do not conduct any training for Point-LLM, thanks to the instruction tuning of ImageBind-LLM. As there is no existing benchmark for 3D instruction models, referring to Vicuna (Vicuna, 2023), we adopt two powerful LLMs, GPT-4 (OpenAI, 2023) and Bard (Google, 2023), for evaluation, and sample 1,000 3D-caption pairs from Cap3D (Luo et al., 2023) as the test set, which is a large-scale 3D object captioning dataset built upon Objaverse (Deitke et al., 2023). Specifically, for two generated responses to compare, we feed them into the evaluator LLM to ask which one is closer to the ground-truth caption, and count the number of 'Win', 'Tie', and 'Lost' for comparison. We regard ImageBind-LLM as a baseline, which takes one rendered image of the point cloud as input for 3D Q&A.

**Analysis.** In Figure 6, we present the results evaluated by GPT-4 and Bard. Compared to the baseline ImageBind-LLM, by using our Point-Bind for 3D encoding, Point-LLM achieves significantly more 'Win'. This fully indicates our Point-Bind aligned with multi-modality can better extract 3D spatial geometries than ImageBind's 2D encoder. If we do not adopt the visual cache model, the 3D question-answering performance would be severely harmed, due to the 2D-3D modality discrepancy between training and inference. In Figure 3, we provide the question-answering examples of Point-LLM, which shows favorable 3D instruction-following and multi-modal reasoning capacity. As shown, for either English or Chinese instructions, Point-LLM can effectively incorporate the

Table 2: **Performance of Text-to-3D Generation.** We report the Fréchet Inception Distance (FID), Fréchet Point Distance (FPD), and CLIP R-Precision (RP) scores.

| Method | FID ($\downarrow$) | FPD ($\downarrow$) | RP ($\uparrow$) |
|---|---|---|---|
| CLIP-Forge | 162.87 | 37.43 | 3.85 |
| GLIDE + DVR | 212.41 | 41.33 | 7.69 |
| LAFITE + DVR | 135.01 | 37.55 | 3.85 |
| ISS | 124.42 | 35.67 | 7.69 |
| **Point-Bind** | **112.25** | **23.06** | **15.39** |

Table 3: **Performance of 3D Zero-shot Classification.** We report the classification accuracy (%) on ModelNet40 (MN40) and ScanObjectNN (ScanObj) datasets.

| Method | Encoder | MN40 | ScanObj |
|---|---|---|---|
| PointCLIP | CLIP | 20.2 | 21.3 |
| ULIP | Point-BERT | 60.4 | 49.9 |
| PointCLIP V2 | CLIP | 64.2 | 50.1 |
| **Point-Bind** | Point-BERT | 76.3 | **61.3** |
| | I2P-MAE | **78.0** | 56.8 |

spatial geometry of input point clouds and generate detailed language responses. It obtains a comprehensive 3D understanding for both global and local characteristics, e.g., recognizing the pattern of the piano keyboard and the shape of the airplane's wing and tail. Then, our Point-LLM can also respond with cross-modal understanding. For an input 3D model with a 2D image or audio, Point-LLM can enable LLaMA to take both two conditions into understanding and reasoning, which thus incorporates multi-modal semantics in the output language response.

## 4.2 Composed 3D Cross-modal Retrieval

**3D Cross-modal Retrieval.**   To evaluate the multi-modal alignment of Point-Bind, we first experiment with several cross-modal retrieval tasks between 3D and another modality, i.e., 2D and text. We evaluate our method on the multi-modal ModelNet40 (Wu et al., 2015) dataset, and obtain the retrieved results by ranking feature similarities. As shown in Table 1, our Point-Bind attains leading performance on all benchmarks compared with prior works (Zhang et al., 2022b; Zhu et al., 2022; Xue et al., 2022; 2023). In particular, for 2D-to-3D and text-to-3D retrieval, Point-Bind surpasses the ULIP (Xue et al., 2022) significantly by **+14.29%** and **+13.99%**, respectively. This indicates the superior cross-modal understanding capacity of our approach.

**Embedding-space Arithmetic.**   With the multi-modal alignment, we further explore the capability of embedding composition, i.e., the embedding-space arithmetic of 3D and other modalities, e.g., audio. We utilize 3D objects from ShapeNet (Chang et al., 2015) and TextANIMAR 2023 (Challenge, 2023), and audio clips from ESC-50 (Piczak, 2015). We simply add the 3D and audio embeddings respectively from Point-Bind and ImageBind, and retrieve 2D images from ImageNet (Deng et al., 2009). In Figure 7, we show the results of 2D image retrieval with the composed embeddings between 3D and audio. As shown in the first row, with the combined embeddings of a 3D dog and sea-wave audio, we effectively retrieve 2D images of dogs by the sea. Similarly, with the combination of a 3D laptop and keyboard-typing audio, the obtained images show someone is working with a laptop, or a cat inadvertently presses on the keyboard. Likewise, the last row retrieves images of bears hunting by the water by using embeddings of a 3D bear and audio of flowing water. The examples demonstrate the 3D features from Point-Bind can be directly added with other aligned modalities, and incorporate their semantics, achieving favorable composed cross-modal retrieval.

## 4.3 Any-to-3D Generation

**Settings.**   We adopt a learnable projection layer to connect the decoder of ISS (Liu et al., 2022) with the embedding space of Point-Bind by single view reconstruction (SVR). For quantitative evaluation, we compare several existing methods (Sanghi et al., 2021; Jain et al., 2022b; Liu et al., 2022) for text-to-3D generation, and adopt three criteria, Fréchet Inception Distance (FID) (Heusel et al., 2017), Fréchet Point Distance (FPD) (Shu et al., 2019), and CLIP R-Precision (RP) (Park et al., 2021). Please refer to Supplementary Material for 3D generation with other modalities and their composition.

**Analysis.**   In Table 2, the quantitative comparison shows our approach achieves lower FID/FPD values, outperforming other methods for both 3D generation quality and shape correspondence. The competitive CLIP R-Precision score also suggests a higher consistency between text and 3D shapes of our method. In Figure 2, we also show several qualitative results of any-to-3D generation and instruction-based editing powered by Point-Bind, e.g., generating 3D meshes from audio and point clouds, along with 3D shape editing by input language instructions (More visualizations are

shown in Supplementary Material). This demonstrates the well-aligned embedding space of 3D and multiple modalities in Point-Bind.

### 4.4 3D ZERO-SHOT UNDERSTANDING

We investigate our open-word understanding ability, i.e., recognizing novel classes, by 3D zero-shot classification on ModelNet40 (Wu et al., 2015) and ScanObjectNN (Uy et al., 2019) datasets.

**Settings.** Following previous CLIP-based works, we utilize the text embeddings from Image-Bind's (Girdhar et al., 2023) text encoder to construct the zero-shot classifier. Specifically, we apply a simple template of *'a [CLASS]'* for the 40/15 categories of ModelNet40/ScanObjectNN, and calculate the cosine similarity between 3D and all textual embeddings, selecting the most similar one as the final prediction. Moreover, as our 3D embeddings are also aligned with the audio modality, our approach also supports the audio-referred 3D zero-shot recognition by regarding audio embeddings of different categories as the classifier.

**Analysis.** We report the 3D zero-shot classification accuracy in Table 3, where our Point-Bind can surpass existing methods (Zhang et al., 2022b; Zhu et al., 2022; Xue et al., 2022; 2023) on both benchmarks. With the same encoder as Point-BERT (Yu et al., 2022), our approach outperforms ULIP by significant margins, **+15.9%** and **+11.4%** accuracy on the two datasets. For audio-referred 3D classification, we select six categories that can make a sound in ModelNet40 for evaluation (airplane, car, guitar, keyboard, piano, toilet), for which our Point-Bind attains 89.3% accuracy, a little worse than the text-referred 91.8%. This indicates the unified representation space of Point-Bind leads to strong emergent 3D open-world recognition.

### 4.5 ABLATION STUDY

In Table 4, we conduct two ablation studies to verify the effectiveness of the multi-modal training and 3D encoder selection in Point-Bind. We report the zero-shot classification accuracy on ModelNet40. In the left part of the table, we progressively add the modality in training data, and observe the performance increasing. This indicates the contrastive supervision from more modalities contributes to a better 3D joint embedding space. In the right

Table 4: **Ablation Study** investigating different training data modality and 3D encoders. We report the zero-shot classification on ModelNet40 dataset.

| Modality of Training Data | | | | Acc. | 3D Encoder | Acc. |
|---|---|---|---|---|---|---|
| Text | 3D | Image | Audio | | | |
| ✓ | ✓ | - | - | 70.43 | PointNeXt | 67.96 |
| - | ✓ | ✓ | - | 68.72 | Point-BERT | 76.30 |
| ✓ | ✓ | ✓ | - | 76.96 | Point-M2AE | 77.47 |
| ✓ | ✓ | ✓ | ✓ | **78.00** | I2P-MAE | **78.00** |

part, we utilize different 3D encoders in Point-Bind, i.e., Point-BERT (Yu et al., 2022), Point-NeXt (Qian et al., 2022a), and I2P-MAE (Zhang et al., 2023a). As reported, the pre-trained Point-BERT and I2P-MAE can achieve much better performance, indicating the importance of 3D pre-training to boost the multi-modal alignment.

## 5 CONCLUSION

In this paper, we propose **Point-Bind**, a general 3D multi-modality model that aligns 3D point clouds with multi-modalities, guided by ImageBind. By aligning 3D objects with their corresponding image-audio-text pairs, Point-Bind obtains a joint embedding space, and exhibits promising 3D multi-modal tasks, such as any-to-3D generation, 3D embedding arithmetic, and 3D open-world understanding. The emergent ability of Point-Bind significantly lowers the bar for efficiently achieving many new cross-modal applications. Upon that, we further introduce **Point-LLM**, a 3D large language model (LLM) with superior instruction-following and multi-modal reasoning capabilities. Extensive experiments have demonstrated the effectiveness and significance of our 3D multi-modal framework. Future work will focus on aligning multi-modality with more diverse 3D data, such as indoor and outdoor scenes, which allows for a wider range of 3D-centric scenarios.

## A OVERVIEW

- Section B: Additional experiments.
- Section C: Related work.
- Section D: Additional implementation details.

## B ADDITIONAL EXPERIMENTS

**Cross-modal Retrieval on More Modalities.** To verify the potential of Point-Bind to align multi-modalities, we conduct cross-modal retrieval between 3D and more modalities, i.e., video, depth, and infrared data. We utilize the following work of ImageBind (Girdhar et al., 2023), Language-Bind (Zhu et al., 2023a), as guidance, and pre-train Point-Bind under the same paradigm. By aligning 3D with the image space of LanguageBind, Point-Bind achieves a unified space with multi-modalities including video, depth, and infrared data. As shown in Figure 8, with the 3D car/person as input, Point-Bind effectively retrieves corresponding video, depth, and infrared data with the same semantics. This indicates the superior cross-modal understanding capacity of our approach.

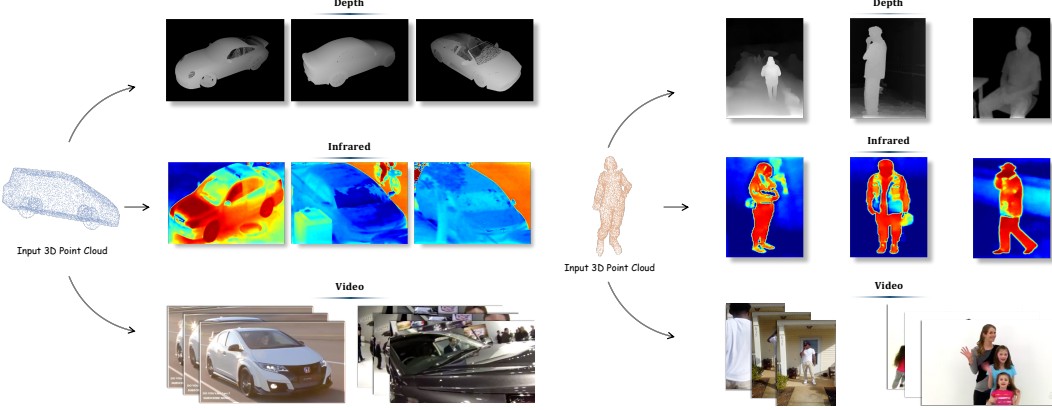

Figure 8: **Additional Visualization of Cross-modal Retrieval.** We visualize the cross-modal retrieval between 3D and three new modalities, i.e., video, depth, and infrared data. Note that, for these modalities, we utilize LanguageBind (Zhu et al., 2023a) as the guidance for pre-training Point-Bind.

**Quantitative Results of Any-to-3D Genration.** Besides text-to-3D generation, we quantitatively demonstrate the efficacy of Point-Bind on any-to-3D generation in Table 5. We generate the 3D mesh conditioned on multi-modalities and their embedding-space arithmetic, i.e., directly combining embeddings from different modalities to guide 3D generation. We adopt different settings for different modalities. For audio-to-mesh generation, we only generate objects of the car, airplane,

and boat categories considering the limited class number. We sample 10 audio clips per category from ESC-50 dataset (Piczak, 2015) as input. The airplane take-off sound, car horn, and sea wave sound are selected to generate the airplane, car, and boat categories, respectively. For image-to-mesh generation, we sample 10 images corresponding to ShapNet's 13 categories from ImageNet dataset (Deng et al., 2009) as the 2D prompt. For point-to-mesh synthesis, we sample 10 point clouds per category from the ShapeNet dataset (Chang et al., 2015) as prompt. Compared to text-to-3D generation, the results in Table 5 suggest that Point-Bind can also achieve satisfactory generation quality with other modalities as conditions.

Table 5: **Quantitative Results of Any-to-3D Generation.** We report the Fréchet Inception Distance (FID) and Fréchet Point Distance (FPD) scores for comparison.

| Source Modality | FID ($\downarrow$) | FPD ($\downarrow$) |
|---|---|---|
| Audio | 166.97 | 30.46 |
| Image | 95.77 | 19.41 |
| Point Cloud | 86.14 | 20.13 |
| Image + Text | 86.79 | 26.13 |
| Point Cloud + Text | 88.78 | 26.39 |
| Point Cloud + Image | 87.03 | 21.19 |

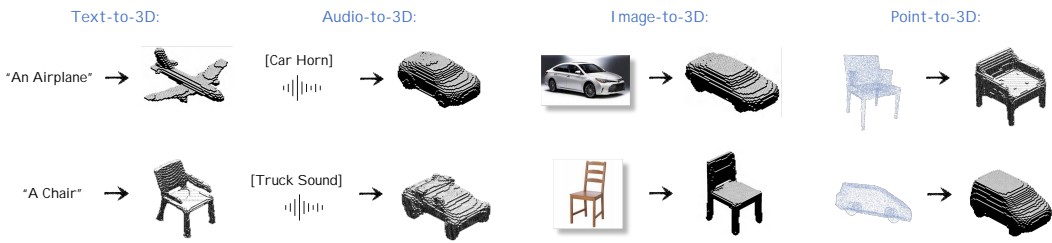

Figure 9: **Any-to-3D Generation** based on CLIP-Forge (Sanghi et al., 2021). Besides ISS (Liu et al., 2022), our Point-Bind is generalized to combine any text-to-3D models for any-to-3D generation.

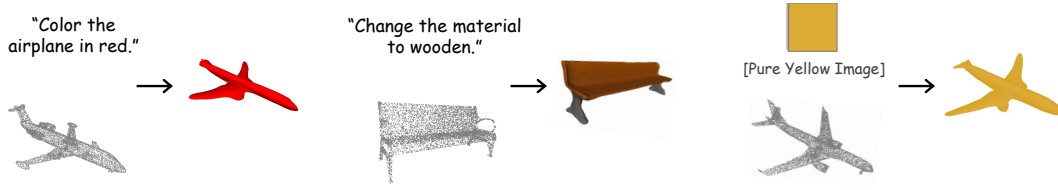

Figure 10: **3D Editing with Multi-modal Instructions.** Within the joint 3D embedding space of Point-Bind, we can effectively edit input 3D point clouds with multi-modal instructions, e.g., language or image.

**Any-to-3D Generation with CLIP-Forge (Sanghi et al., 2021).** Besides ISS (Liu et al., 2022), we also adopt the decoder of CLIP-Forge and show the examples of any-to-3D generation powered by Point-Bind in Figure 9. For text, audio, and point cloud prompts, our approach can all produce satisfactory 3D meshes. This demonstrates that Point-Bind generalizes well and can guide other 3D generation models conditioned on multi-modalities.

**3D Editing with Multi-modal Instructions.** Besides the any-to-3D generation, our approach can further enable 3D editing with multi-modal instructions, as visualized in Figure 10. For example, given a 3D airplane, we can provide a language instruction, "Color the 3D shape in red", or a pure yellow picture as the visual instruction. Then, we respectively feed them into Point-Bind's 3D encoder and ImageBind's text or image encoder. Due to the joint embedding space, the generative decoder can incorporate their semantics and output the airplane in red/yellow. Likewise, given an ordinary 3D bench, we can provide instructions like "Modify the material to wooden". The model can correspondingly generate a wooden chair. Therefore, benefiting from the emergent capacity of Point-Bind, we can simply achieve any-to-3D generation and editing, exhibiting favorable training efficiency and generalization capability.

**Additional Comparison and Analysis with ULIP.** The teacher model of Point-Bind, Image-Bind (Girdhar et al., 2023), has different pre-training settings with ULIP's (Xue et al., 2022) teacher model, SLIP (Mu et al., 2021). In this paragraph, we compare Point-Bind and ULIP with the same pre-trained teacher models. We first reproduce a ULIP model also pre-trained by CLIP's ViT-H image encoder, which is the same as ImageBind's image encoder. Note that, ImageBind freezes the ViT-H image encoder and text encoder of OpenCLIP during its pre-training. That is, ImageBind and OpenCLIP share the same weights in their image and text encoders. As shown in Table 6, for zero-shot classification on ModelNet40 (Wu et al., 2015), although the ULIP's performance can be improved by the ViT-H image encoder, our approach still performs better via a joint multi-modal embedding space.

**Generalizability of Point-Bind with Techniques from JM3D (Wang et al., 2023a).** JM3D (Wang et al., 2023a) proposes two delicate approaches to enhance the multi-modal pre-training of 3D models: Structured Multimodal Organizer (SMO) and Joint Multi-modal Alignment (JMA). SMC adopts multi-view rendered images and hierarchical text for more comprehensive rep-

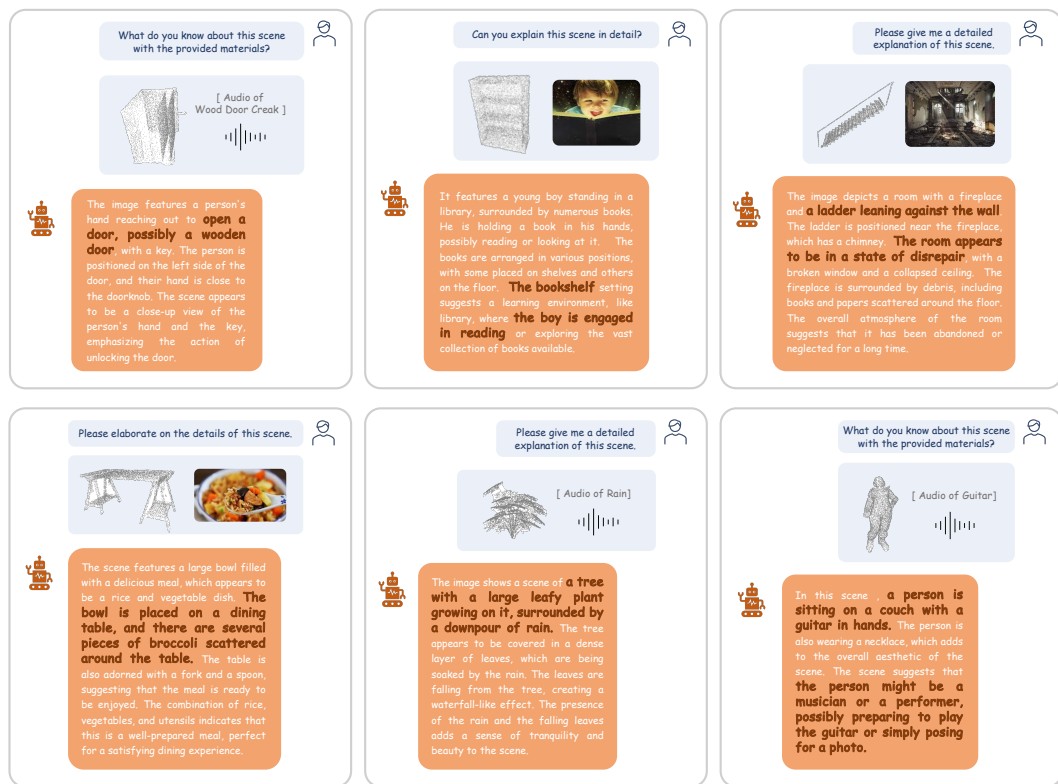

Figure 11: **Additional 3D Question-answering Examples of Point-LLM.** Point-LLM can effectively generate detailed responses and conduct superior cross-modal reasoning, based on the given multi-modal instructions.

Table 6: **Comparison to ULIP by Teacher Models with The Same Image Encoder: ViT-H.**

| Method | Teacher Model | Image Encoder | Accuracy |
|---|---|---|---|
| ULIP | OpenCLIP (Ilharco et al., 2021) | ViT-L | 60.4% |
| ULIP | OpenCLIP (Ilharco et al., 2021) | ViT-H | 73.2% |
| Point-Bind | ImageBind (Girdhar et al., 2023) | ViT-H | **76.3%** |

resentation, and JMA aims to achieve better mult-modal synergy by generating joint vision-language features. We also add the two techniques in JM3D into our Point-Bind for the image and text modalities within ImageBind (Girdhar et al., 2023), and evaluate on two benchmarks: 3D zero-shot classification and cross-modal retrieval on ModelNet40 (Wu et al., 2015). As shown in Table 7, the capabilities of Point-Bind are well enhanced by integrating SMO and JMA, indicating the importance of more comprehensive vision-language guidance.

**Generalizability of Point-Bind with Techniques from CG3D (Hegde et al., 2023).** CG3D (Hegde et al., 2023) shares a similar contrastive learning paradigm with ULIP, and introduces learnable visual prompts for CLIP's image encoder for better adaption of 2D rendered images. For our Point-Bind, we also add learnable visual prompts to the image encoder of ImageBind, and report the results in Table 7. On both benchmarks, the prompting approach from CG3D can improve the performance of Point-Bind, which demonstrates the effectiveness of fine-tuning the pre-trained image embeddings.

**Additional 3D Question-answering Examples.** We provide more 3D question-answering examples in Figure 11, showing the 3D instruction-following and multi-modal reasoning capacity of

Table 7: **Performance(%) of Point-Bind with JM3D (Wang et al., 2023a) and CG3D (Wang et al., 2023a) on 3D Zero-shot Classification and Cross-modal Retrieval Tasks.**

| Method | Zero-shot Cls. | 3D → 3D | 2D → 3D | 3D → 2D | Text → 3D |
|---|---|---|---|---|---|
| Point-Bind | 78.0 | 63.2 | 34.6 | 42.8 | 64.5 |
| Point-Bind w JM3D | **78.4** | **64.1** | **35.5** | **43.9** | 64.7 |
| Point-Bind w CG3D | 78.2 | 63.5 | 34.3 | 43.2 | **64.8** |

Point-LLM. As shown, given a 3D shape with a 2D image or audio, Point-LLM effectively enables LLaMA (Touvron et al., 2023) injected with multi-modal semantics, and responds with cross-modal understanding and reasoning. Additionally, as shown in Figure 12, we show more examples of Point-LLM for straightforward question answering, e.g., "How to start it?", "What is the purpose of this thing?". Our model can respond with precise answers that correspond to the input point cloud.

**Examples of Indoor Scene Understanding.** We further implement a scene-level variant of our model, termed Point-LLM$_{Scene}$. We focus on the understanding of indoor scenes on ScanNet (Dai et al., 2017), and show the qualitative examples in Figure 13. Specifically, to obtain the scene-level understanding capacity, we fine-tune our object-level Point-LLM by an existing 3D question-answering dataset (Wang et al., 2023c) constructed from ScanRefer (Chen et al., 2020a). We add three MLP layers with residual connections between Point-Bind's 3D encoder and the LLM, which is responsible for learning the scene-level 3D geometries. We only enable the new MLP layers to be trainable, while keeping other components frozen to preserve the pre-trained cross-modal knowledge. As shown, our model can respond with detailed and reasonable answers that correspond to the input 3D scene and target object.

## C  RELATED WORK

**Multi-modality Learning.** Compared to single-modal approaches, multi-modal learning aims to learn from multiple modalities simultaneously, achieving more robust and diverse representation learning. Numerous studies have proved its efficacy, involving 2D images, videos, texts, and audio (Desai & Johnson, 2021; Fang et al., 2021; Nagrani et al., 2022), and enhance the cross-modal performance for downstream tasks (Lin et al., 2021b; Ramesh et al., 2021; Botach et al., 2022; Guo et al., 2023c), and video-text-audio integration for text generation (Lin et al., 2021a). The representative vision-language pre-training, CLIP (Radford et al., 2021), effectively bridges the gap between 2D images and texts, which encourages further exploration of cross-modality learning. Recently, ImageBind (Girdhar et al., 2023) successfully aligns six modalities in a joint embedding space, unleashing the power for emergent zero-shot cross-modal capabilities. However, ImageBind fails to investigate its efficacy on 3D point clouds. In the 3D domain, most existing cross-modal works introduce vision-language alignment (Zhang et al., 2022b; Xue et al., 2022; Afham et al., 2022; Guo et al., 2023a; Chen et al., 2023a) into 3D point clouds, and mainly focus on open-world recognition tasks, which ignore the potential of multi-modal semantics for wider 3D applications. In this paper, our Point-Bind develops a general 3D multi-modality model that aligns 3D point clouds with six other modalities guided by ImageBind, allowing for more diverse 3D cross-modal understanding.

**Large Models in 3D.** Large-scale pre-trained models have achieved remarkable downstream performance in language and 2D image processing. Inspired by this, many efforts have introduced 2D and language large models, to assist in 3D learning. The prior PointCLIP series (Zhang et al., 2022b; Zhu et al., 2022; Huang et al., 2022) project 3D point clouds into depth maps, and utilize CLIP (Radford et al., 2021) for zero-shot recognition. Image2Point (Xu et al., 2022) instead converts 2D pre-trained models into 3D space as a good network initialization. By contrastive learning, ULIP series (Xue et al., 2022; 2023) and other works (Liu et al., 2023a; Hegde et al., 2023) pre-train 3D networks guided by the vision-language embedding space of CLIP. Another branch of work employs CLIP to guide the text-conditioned generation of 3D objects (Jain et al., 2022a; Sanghi et al., 2021; Xu et al., 2023a; Liu et al., 2023b) or stylized meshes (Mohammad Khalid et al., 2022; Michel et al., 2021) by encoding descriptive textual input. Some works also adopt GPT-3 (Brown et al., 2020) to enhance the language-based understanding of 3D spatial geometry, such as Point-CLIP V2 (Zhu et al., 2022) and ViewRefer (Guo et al., 2023b). Different from them, we utilize

| Input 3D Object | Question | Answer of Point-LLM |
|---|---|---|

To start the car, you would **typically press the accelerator pedal down while holding the brake pedal down simultaneously.** This will engage the car's starter motor, which will turn the engine over, and the car will start to move.

*How to start it?*

The purpose of this thing is to **transport passengers from one location to another.** In this scene, a white and black jet airplane is on the runway, preparing to take off. It is a large commercial airplane, which means it is designed to carry a significant number of passengers and their luggage. The airplane **serves as a mode of transportation for people to travel long distances quickly and efficiently, connecting various cities and countries around the world.**

*What is the purpose of this thing?*

To use the bathtub, you would **first fill it with water to the desired level. Once the tub is filled, you can sit or lay down in it, taking care to avoid splashing water outside the tub.** The bathtub is designed to provide a comfortable and relaxing experience, allowing you to soak in the warm water and enjoy the hydrotherapy benefits. It is essential to ensure that the water level is not too high, as it can pose a risk of slipping or falling. Additionally, it is important to keep the area around the bathtub clean and dry to prevent any potential damage or accidents.

*How to use it?*

There are several examples of real-life objects that are shaped like a cone or pyramid. **Some examples include: 1. Pyramids:** The most famous example is the Pyramids of Giza, which are triangular pyramids built by the ancient Egyptians as tombs for their pharaohs. **2. Volcanoes:** Volcanoes, especially those that have recently erupted, can have a cone-like shape. The cone is formed as lava or pyroclastic cone material is ejected from the volcano's vent. **3. Mountains:** Many mountains have a pyramid-like shape, with a wide base and a narrow peak. **Examples include Mount Fuji in Japan, Mount Kilimanjaro in Tanzania, and the Pyramidion on the Pyramid of Giza.**

*What thing in real life is shaped like this? Please give me some examples.*

This is **a small side table or nightstand, typically used to hold items such as a bedside lamp, a phone, a glass of water, or a small book.** You can decorate it by placing a small tray or a decorative plate on top of it to hold items. You can also use a decorative box or a small shelf to store additional items. In the image, the nightstand is currently empty, so **you could add a decorative lamp, a small vase with flowers, or other decorative items to enhance its appearance.**

*What is this? What do people usually put on it? How to decorate it?*

**One famous classical work involving a grand piano is Sergei Rachmaninoff's "Piano Concerto No. 2 in C minor, Op. 18."** This piece is considered a masterpiece of the Romantic era, widely recognized for its emotional depth, technical challenges, and the piano's prominent role in the composition. **Another famous work is Maurice Ravel's "Gaspard de la nuit,"** a piano suite that features three movements, each with its own unique character and musical style. The suite is based on a story by French writer Aloysius Bertrand and is known for its imaginative nature, with the third movement, "Ballade," being particularly well-known for its haunting melody and harmonies.

*Provide some famous classic works concerning this object.*

Figure 12: **Examples of Point-LLM for Straightforward Question Answering.**

ImageBind (Girdhar et al., 2023) to construct a joint embedding space between 3D point clouds and multiple modalities. The derived Point-Bind can well leverage the multi-modal semantics for general 3D cross-modal understanding, generation, and question answering. There are a couple of very recent efforts that introduce LLMs into 3D, concurrent to our Point-LLM. Different from us, they either project 3D data into multi-view images for encoding (Hong et al., 2023b), or require large-

**Input 3D Scene**        **Instruction & Response of Point-LLM**

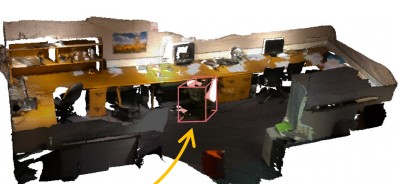

**Instruction:**

What is the target object? Where is its exact location?
Share your thoughts on the room's overall design and ambiance.

**Response:**

The object in question is **a white monitor placed on top of a desk.** It is located on the left of another monitor and to the right of a telephone. The monitor has a rectangular shape and is positioned on the left side of the desk.

The fact that the monitor is placed on top of a desk indicates its intended use as a workspace or study area. The presence of a telephone nearby suggests that the monitor may be used for communication or multitasking, allowing the user to easily access both the monitor and phone while working.

The surrounding objects, such as the keyboard, monitor, desk, cabinet, pillow, wall, bookshelf, and other miscellaneous objects, contribute to the overall functionality and organization of the workspace. **The fact that the monitor is placed on the left of another monitor and to the right of the telephone suggests that the area is specifically designed for multitasking or work-related activities that require a combination of visual and auditory input.**

Overall, the white monitor on the desk plays an important role in the functionality and productivity of the workspace, providing a reliable and efficient means of accessing information, data, or communication tools.

**Instruction:**

Inspect the target object thoroughly and describe its details.

**Response:**

The object in question is **a black metal chair placed at the center of the room. Its placement in the middle suggests a central location within the space. Positioned next to a desk with a monitor,** the chair serves as a seating option for someone working or studying at the desk. Its proximity to the desk also implies that it is likely used for a work or study purpose.

The presence of multiple other chairs in the room suggests that this is a shared space where multiple individuals may work or study. The chairs are arranged in a symmetrical pattern, creating a balanced and organized seating arrangement.

The presence of **other objects, such as a desk, keyboard, computer, and wall,** further indicates that this area serves as an office or workspace. **The placement of the chair in front of the monitor and its proximity to the table indicate that it is likely used for computer work.**

Overall, this setup suggests a well-organized and functional space for work or study. The arrangement of the chairs and other objects creates a comfortable and visually pleasing environment for individuals working or studying in the area.

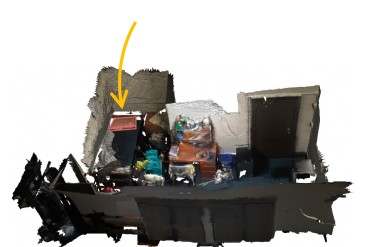

**Instruction:**

Elaborate on the displayed target object with great details.
What items are around this object?

**Response:**

The described object is **a rectangular white pillow placed on a bed. Positioned on the left side of a bookshelf,** the pillow adds visual appeal and comfort to the surrounding area. The placement on the bed is appropriate, allowing the pillow to be used to rest people, and complement the overall interior design of the room or space.

In the 3D scene, the described pillow is **surrounded by multiple other objects, such as books, a radiator, a backpack, a desk, and a chair. These items suggest that the scene is a living or working space,** where books and desk-related objects are commonly found.

The pillow provides a comfortable seating option for users, and as a decorative element, adding character and warmth to the area while complementing the surrounding objects in the room. It creates a cozy and inviting atmosphere for people who interact with the space, whether for relaxation or work purposes.

Figure 13: **3D Scene Understanding Capability of Point-LLM$_{Scene}$.**

scale 3D instruction data for fine-tuning (Xu et al., 2023b; Wang et al., 2023b). More importantly, they cannot generate responses conditioned on both 3D and multi-modal input. Thanks to the joint embedding space of Point-Bind, our Point-LLM can discard the expensive 3D instruction tuning, and respond via 3D multi-modal reasoning.

**Pre-training in 3D.** In recent years, significant progress has been made in supervised learning for 3D vision tasks (Qi et al., 2016; 2017; Qian et al., 2022a; Zhang et al., 2023b; Zhu et al., 2023b). However, these approaches lack satisfactory generalization capabilities for out-of-domain data. To address this, self-supervised learning has emerged as a promising solution to enhance

3D transfer learning (Chen et al., 2023a; Yu et al., 2022; Li et al., 2019; Poursaeed et al., 2020). Most self-supervised pre-training methods employ an encoder-decoder framework to encode point clouds into latent representations and then reconstruct the original data form (Sauder & Sievers, 2019; Wang et al., 2021; Rao et al., 2020). Therein, Point-MAE (Pang et al., 2022) and Point-M2AE (Zhang et al., 2022a) introduce masked autoencoders (He et al., 2021) into 3D point clouds pre-training, achieving competitive results on different 3D tasks. Alternatively, cross-modal pre-training approaches are also leveraged to enhance the 3D generalization ability (Wang et al., 2022; Qian et al., 2022b; Liu et al., 2021a; Qi et al., 2023). For example, ACT (Dong et al., 2022) and I2P-MAE (Zhang et al., 2023a) utilize pre-trained 2D transformers as teachers to guide 3D representation learning. Inspired by previous works, we adopt collected 3D-image-text-audio pairs for self-supervised pre-training, and regard ImageBind's encoders as guidance for contrastive learning. In this way, the Point-Bind is pre-trained to obtain a joint embedding space between 3D and multi-modality, allowing for superior performance on different 3D downstream tasks.

## D  ADDITIONAL IMPLEMENTATION DETAILS

**Multi-modal Training of Point-Bind.**   To align 3D with multi-modalities, we adopt a pre-trained I2P-MAE (Zhang et al., 2023a) as the 3D encoder of Point-Bind by default, and utilize the collected 3D-image-text-audio pairs for pre-training. We utilize a pre-trained ImageBind (Girdhar et al., 2023) with a ViT-H (Dosovitskiy et al., 2020) image encoder. We only update the 3D encoder with the newly added projection network, and freeze the encoders of other modalities in ImageBind. The projection network is composed of two linear layers with an intermediate LayerNorm (Ba et al., 2016). We train Point-Bind for 300 epochs with a batch size of 64, and adopt AdamW (Loshchilov & Hutter, 2017) as the optimizer with a learning rate of 0.003.

**3D Cross-modal Retrieval.**   We utilize ModelNet40 (Wu et al., 2015) to evaluate Point-Bind on cross-modal retrieval tasks without training. The test set of ModelNet40 provides 2,468 samples with two modalities, i.e., 2D images rendered from 3D meshes and corresponding 3D point clouds. We adopt the Mean Average Precision (mAP) score as the criterion, which measures whether the retrieved data belongs to the same class as the query data. We encode 3D point clouds with Point-Bind and conduct four cross-modal retrieval tasks, i.e., 3D-to-3D, 2D-to-3D, 3D-to-2D, and text-to-3D retrieval. For the text prompt, we adopt and separately encoder 64 prompt templates in ULIP (Xue et al., 2022) on each category, and average them as the text embeddings. For the 2D image prompt, we follow (Jing et al., 2021) to utilize multi-view images where the view number is $\in$ $\{1, 2, 4\}$. We average the performance under the three view settings as the final result.

**Any-to-3D Generation.**   We adopt Image as Stepping Stone (ISS) (Liu et al., 2022) to verify Point-Bind's ability of multi-modal feature alignment. We first optimize a projection layer that transfers Point-Bind image features to ISS 3D shape space. Then, we generate 3D shapes from text features based on the pre-trained projection layer and ISS decoder. The ShapeNet (V2) dataset(Chang et al., 2015) with 13 object categories is utilized to train the model. We follow ISS and adopt a text description set with four texts per category. To demonstrate 3D generation quality, we adopt FID, FPD, and CLIP R-precision as criteria. FID reflects the quality of rendered 2D images from generated 3D shapes. FPD measures the quality of point clouds extracted from generated shapes based on a pre-trained PointNet model (Qi et al., 2016) following ISS. Additionally, we further adopt CLIP R-precision to evaluate the consistency between the text inputs and generated shapes. We build a text description set, which contains our description prompts and 234 additional texts from CLIP-Forge (Sanghi et al., 2021). Then, we perform per-shape CLIP-R-Precision to retrieve the right description for each generated shape and calculate the retrieval accuracy. To give a comprehensive comparison, we mainly compare our approach to three text-to-mesh generation models, CLIP-Forge (Sanghi et al., 2021), Dream Fields (Jain et al., 2022b), and ISS. Note that Dream Fields can not synthesize 3D shapes directly, so we do not need to evaluate its FPD metric. In addition, two baselines, GLIDE/LAFITE+DVR, which first create images and then generate 3D meshes are also included. Following ISS, we first use GLIDE (Nichol et al., 2021) or LAFITE (Zhou et al., 2022) to create 2D images and then generate 3D shapes via DVR (Niemeyer et al., 2020).

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
