# OpenReview forum: "Point-Bind & Point-LLM: Aligning Point Cloud with Multi-modality for 3D Understanding, Generation, and Instruction Following"
_ICLR.cc/2024/Conference — ICLR 2024 Conference Desk Rejected Submission_

### Official Review · Reviewer_Fjjw · 2023-10-17

**Soundness:** 3 good
**Presentation:** 3 good
**Contribution:** 2 fair
**Rating:** 6
**Confidence:** 4

**Summary:**

This paper presents Point-Bind, a method that aligns 3D point clouds with other modalities such as images, language, and audio to learn a joint embedding space. By leveraging this learned space, various multi-modal applications including 3D QA/captioning, generation, retrieval, and zero-shot recognition are introduced. Quantitative experiments and qualitative illustrations demonstrate its effectiveness.

**Strengths:**

1. The research topic is of important.
2. The paper's structure is clear and easy to follow.
3. Extensive experiments and illustrations demonstrate the promising properties of the joint feature space that has been learned.

**Weaknesses:**

1. Aligning point clouds with other modalities, such as images and text, using contrastive learning is not new in the 3D point cloud field. This paradigm has already been explored by previous ULIP (CVPR 2023). The main difference is that ULIP aligns with the CLIP variant, while this paper aligns with ImageBind, which includes an additional audio modality.

2. From my perspective, the introduced Point-LLM seems more like a better utilization of ImageBind-LLM with visual cache. This is similar to learning a projection layer to utilize ISS for 3D generalization, rather than proposing a separate model.

3. Some conclusions are drawn from experimental comparisons that may not be adequately fair.
    - In Figure 6, ImageBind-LLM takes a single rendered image of the point cloud as input, while Point-LLM takes the raw point cloud itself (please correct me if I am mistaken). However, it seems that the training image-text pairs of ImageBind-LLM hardly include single-view rendered images of CAD models. On the other hand, Point-LLM directly uses point clouds during its training process. Therefore, this might not be fair.  The authors may present results where Point-LLM also uses rendered images as input or feeds multi-view rendered images into ImageBind-LLM. In addition, it would be more illustrative if the authors could provide such a comparison with the concurrent [PointLLM](https://github.com/OpenRobotLab/PointLLM).
    - When comparing Table 3 with Table 1, the results of ULIP-2 are missing in Table 1.
    - In Tables 1 and 3, both Point-Bind and ULIP utilize Point-BERT as the 3D encoder. However, they differ in terms of the architecture of the image/text encoder and the training data utilized. For example, ULIP aligns with SLIP (ECCV 2022), which incorporates ViT-L and is trained with 15M data. On the other hand, Point-Bind aligns with ImageBind, which utilizes ViT-H and is trained with billions of pairs.

**Questions:**

1. In **3D-audio Pairs** of Section 2.2, the authors obtained 9 categories of 3D point clouds paired with extensive audio clips. It would be more informative if the authors could provide information on which 9 categories the audio clips were obtained from and how many of them.
2. The ImageBind does not require the training dataset that pairs all six modalities. However, the authors have collected a unified 3D-image-audio-text dataset. Table 4 indicates that only Text+3D does not perform well. Can the authors provide any analysis? Furthermore, the inclusion of the image modality greatly enhances performance as shown in Table 4. Therefore, I am curious about the results for Image+3D, if available.
3. In **Embedding-space Arithmetic** of Section 4.2, Why not use both 3D and audio embeddings from Point-Bind but "add the 3D and audio embeddings respectively from Point-Bind and ImageBind"?
4. In **Settings** of Section 4.4, why not use 64 templates like in equation (2) but "a simple template of ‘a [CLASS]’ for the 40/15 categories"?
5. Did the authors themselves obtain the results of zero-shot on ScanObjectNN for ULIP and ULIP-2, as shown in Table 3? Is there any analysis explaining why ULIP outperforms ULIP-2?

---

> ### Author Response · Authors · 2023-11-21
> **Response to Reviewer Fjjw (Part #1)**
>
> We sincerely appreciate your detailed and insightful reviews. We hope our response can address your concerns.
>
> ---
> > **Q1: Aligning 3D with pre-trained models using a contrastive paradigm has been a well-used technique, e.g., ULIP**
>
> >
> Sorry for the confusion caused. ***We didn't claim the contrastive paradigm as a contribution in the paper.*** Indeed, we are inspired by previous works to adopt the contrastive paradigm for pre-training, but the main contributions of Point-Bind are in two aspects as follows.
>
> 1. **A 3D multi-modal embedding space.** For the first time, we align 3D with more than two modalities into a joint embedding space. Previous works only verify that 3D can be connected with 2D images and language from CLIP, while our Point-Bind indicates the potential for more modalities including ***audio***, ***video***, ***depth***, and ***infrared data***. As shown in ***Figure 8 of Appendix***, we add the visualization of cross-modal retrieval between 3D and three new modalities (video, depth, and infrared data). This unified space is expected to expand the scope of 3D models to wider cross-modal scenarios.
>
> 2. **Emergent new applications without specific training.** Benefiting from the joint space, our Point-Bind naturally motivates several new cross-modal applications, which ***does not need domain-specific training***. For example, previously, to achieve question answering (QA) with 3D objects, one has to collect large-scale 3D data with paired QA, and spend many computation resources for training. Instead, with our Point-Bind, the 3D QA capability is totally emergent from the joint embedding space given an off-the-shelf 2D QA model, avoiding the expensive data collection and training (referring to our Point-LLM ***in the following Q4***). Likewise, we do not need specific 3D-audio training to achieve audio-referred 3D zero-shot classification, or specific 3D-video training for video-to-3D retrieval. Such an emergent ability of Point-Bind significantly lowers the bar for efficiently achieving many new cross-modal applications.
>
> Therefore, Point-Bind as the first work develops a joint embedding space for 3D multi-modality, and enables many emergent applications without domain-specific training. Besides, we also construct a 3D-image-audio-text paired dataset and introduce a triplet contrastive loss for pre-training, which can be viewed as a miner contribution.
>
>
> ---
> > **Q2: Point-LLM seems more like a better utilization of ImageBind-LLM with visual cache, rather than proposing a separate model**
>
> >
> ***Our motivation is not to develop a new separate model***. Instead, it is exactly ***our strength*** to directly leverage existing off-the-shelf models (ImageBind-LLM) for new 3D-centric applications (3D instruction following), which is powered by the ***naturally emergent capability of Point-Bind***. With the joint embedding space of Point-Bind, our Point-LLM can efficiently achieve 3D instruction-following capacity based on ImageBind-LLM, without the need of expensive 3D-language data collection and training. This brings new insights into 3D domains by exploring an important question: **How to align 3D with LLMs when lack of 3D question-answering data?**
>
> For 2D visual LLMs, e.g., LLaVA [1] and MiniGPT-4 [2], they need extensive image-caption pre-training data to align 2D encoders with LLMs and conduct 2D instruction tuning. However, 3D domains still lack large-scale and high-quality 3D-language datasets. Fortunately, our ***Point-Bind presents a joint embedding space between 3D and multi-modality.*** This allows us to propose an alternative solution to convert the challenging 3D-language alignment into easier 2D-language alignment. Based on the 2D-language alignment of ImageBind-LLM, we can replace its 2D encoder with Point-Bind's 3D encoder, along with a visual cache model to alleviate the 2D-3D domain gap. By regarding 2D as an intermediary, our method can overcome the paucity issue of 3D-caption or 3D question-answering data, which poses a new insight for future 3D LLM research.
>
> Therefore, other than proposing a new model, we aim to demonstrate: the emergent capacity of Point-Bind can help us to ***lower the bar for efficiently*** achieving new 3D-centric cross-modal applications, such as Point-LLM.
>
> Therefore, ***our motivation is not to develop a new separate model***, but to demonstrate: the emergent capacity of Point-Bind can help us to lower the bar for ***efficiently*** achieving many 3D-centric cross-modal applications, such as Point-LLM.
>
> #### Reference:
> #### [1] Visual Instruction Tuning. NeurIPS 2023.
> #### [2] Minigpt-4: Enhancing vision-language understanding with advanced large language models. arxiv 2023.

---

> ### Author Response · Authors · 2023-11-21
> **Response to Reviewer Fjjw (Part #2)**
>
> ---
> > **Q3: Learning a projection layer to utilize ISS for 3D generation is also not proposing a separate model**
>
> >
> As analyzed above in Q1 and Q2, our motivation is to indicate that, with the joint embedding space of Point-Bind, we can efficiently achieve new 3D tasks, ***without expensive task-specific training***. For any-to-3D generation, we only need to learn a lightweight linear projection layer for alignment. ***Such efficient utilization of off-the-shelf models is exactly our advantage***, verifying the emergent capacity of Point-Bind.
>
> Besides, the any-to-3D generation shown in the original submission, our approach can further enable ***3D editing with multi-modal instructions***, as visualized in ***Figure 10 of Appendix***. For example, given a **3D airplane**, we can provide a **language instruction**, "Color the 3D shape in red", or a pure yellow picture as the **visual instruction**. Then, we respectively feed them into Point-Bind's 3D encoder and ImageBind's text or image encoder. Due to the joint embedding space, the generative decoder can incorporate their semantics and output the **airplane in red/yellow**. Likewise, given an ordinary 3D bench, we can provide instructions like "Modify the material to wooden". The model can correspondingly generate a wooden chair.
>
> Therefore, benefiting from the emergent capacity of Point-Bind, we can simply achieve any-to-3D generation and editing with only a lightweight linear projection layer, exhibiting favorable training efficiency and generalization capability.
>
>
> ---
> > **Q4: Compring to ImageBind-LLM with single-view rendered images may not be adequately fair**
>
> >
> Thanks for your advice! To make a more fair evaluation, we respectively compare our Point-LLM with the multi-view ImageBind-LLM and PointLLM (Xu et al., 2023) as follows.
>
> 1. **ImageBind-LLM** can not take raw point clouds as input, so we try our best to transform them into a data form that ImageBind-LLM can process, i.e., rendered 2D images. As our Point-LLM using a 3D encoder cannot encode 2D images, we feed multi-view rendered images into ImageBind-LLM for GPT-4 evaluation, as you suggested. As shown in the table below, with the view number increasing, ImageBind-LLM achieves better 3D captioning capability by understanding more comprehensive 3D geometries. However, our Point-LLM still performs the best by directly encoding point clouds in 3D space with Point-Bind, indicating the superiority of our approach for 3D instruction following.
> | View Number | 1 | 2 | 4 | 8 |
> | ------------ | -------- | -------- | -------- | --------- |
> |Tie |  1.6% | 3.1% | 5.7% | 5.1%|
> | ImageBind-LLM Wins | 34.0% | 35.1% | 35.6% | 39.7% |
> | Point-LLM Wins | **64.4%** | 61.8% | 58.7% | 55.2% |
>
>
> 2. **PointLLM (Xu et al., 2023)** requires a two-stage training strategy to inject 3D semantics into LLMs. After the pre-training stage for 3D-language alignment, PointLLM (Xu et al., 2023) collects a high-quality 3D instruction dataset for specific tuning.
> To fairly compare with it, we also fine-tune our Point-LLM with the same 3D instruction dataset, and show the comparison results in the table below, where our model showcases better 3D captioning accuracy. This demonstrates our multi-modal embedding space of Point-Bind can enhance the 3D understanding performance of LLMs.
> | Method | Rate |
> | ------------ | -------- |
> | PointLLM (Xu et al., 2023) Win | 44.8% |
> | Point-LLM Win | **47.9%** |
> | Tie | 7.3% |
>
>
>
> ---
> > **Q5: ULIP-2 is missing in Table 1**
>
> >
> Thanks for pointing out! We show the ULIP-2 performance in the table below. Note that ULIP-2 is pre-trained by a larger-scale dataset, Objaverse of 800K 3D shapes, while our Point-Bind follows ULIP to pre-train on ShapeNet of only 53K samples. Our approach can outperform ULIP-2 on three cross-modal retrieval results.
>
> ---
> > **Q6: The teacher models of Point-Bind and ULIP are different in architectures and pre-training data**
>
> >
> Yes, it is true that our teacher model, ImageBind, has different pre-training settings with ULIP’s teacher model, SLIP. Here, we reproduce a ULIP model also pre-trained by ***CLIP’s ViT-H image encoder***, which is the same as ImageBind’s image encoder (ImageBind freezes CLIP’s image and text encoders for training other modalities). As shown in the table below, for zero-shot classification on ModelNet40, although the ULIP’s performance can be improved by the ViT-H image encoder, our approach still performs better via a joint multi-modal embedding space.
> | Method         | Image Encoder    | Accuracy |
> | -------------- | ---------- | ---- |
> | ULIP           | ViT-L | 60.4 |
> | ULIP           | ViT-H | 73.2 |
> | Point-Bind | ViT-H    | **78.0** |

---

> ### Author Response · Authors · 2023-11-21
> **Response to Reviewer Fjjw (Part #3)**
>
> ---
> > **Q7: Detailed information of the nine categories paired with audio clips**
>
> >
> Thanks for your advice! We sample nine categories from ESC-50: ***'airplane', 'chirping_birds', 'can_opening', 'car_horn', 'clock_tick', 'keyboard_typing', 'crackling_fire', 'train', 'washing_machine'.*** Each category contains 40 audio samples, with a total number of 360. During training, for a point cloud within the nine categories, we randomly sample an audio sample and adopt data augmentation, e.g., random cropping and volume perturbation, for more robust training.
>
>
> ---
> > **Q7: Compared to ImageBind, why doesn’t ‘Text+3D’ perform well? How about the ‘Image+3D’ result?**
>
> >
> This is mainly because of the ***pre-training data scale***. ImageBind only needs the paired data between image and another one modality, since there are sufficient image-paired data to ensure a robust cross-modal alignment, e.g., 2M for image-audio data and 510K for image-IMU data. In contrast, the 3D-paired dataset is much smaller in scale, i.e., 52.5K of ShapeNet [3]. Therefore, using all four modalities (Text+Image+3D+Audio) provides more contrastive supervision signals, which can to some extent alleviate the influence of data scale.
>
> As you suggested, we also show the ‘Image+3D’ result as follows, where using more modalities for pre-training can achieve better performance.
> |Text                 | 3D   | Image | Audio |    Acc. |
> | ---- | ----- | ----- | -------- | -------------- |
> | ✓  | ✓ | -    | -     | 70.43 |
> | -  | ✓ | ✓ | - | 68.72 |
> | ✓  | ✓ | ✓ | -    | 76.96 |
> | ✓  | ✓ | ✓ | ✓ | **78.00** |
>
>
> ---
> > **Q8: In Embedding-space Arithmetic, why not use both 3D and audio embeddings from Point-Bind?**
>
> >
> Thanks for pointing out! It is a small typo. We have rectified it in the revised manuscript.
>
> ---
> > **Q9: Why not use 64 templates for the 40/15 categories?**
>
> >
> Thanks for pointing out! It is also a typo. We also adopt the 64 templates for zero-shot classification on ModelNet and ScanObjectNN.
>
>
> ---
> > **Q10: Why does ULIP outperform ULIP-2 on ScanObjectNN? Did the authors themselves obtain the results?**
>
> >
> Yes, we utilize the official checkpoint of ULIP-2 to test the zero-shot ScanObjectNN performance, since ***its original paper [4] does not report this result***. For ULIP, we report the numbers referring to its official paper [4].
>
> We think the reason is because of the different pre-training datasets.
> ULIP and our Point-Bind are both pre-trained by ***ShapeNet*** of 53K point clouds, while ULIP-2 is pre-trained by a much larger ***Objaverse*** of 800K 3D samples.
> Normally, larger-scale datasets would lead to better performance, as shown by the ablation study on ***ModelNet40*** of ULIP-2’s paper (Table 5). However, due to the semantic gap between different test datasets, the zero-shot performance of ULIP-2 on ***ScanObjectNN*** exhibits different patterns to that on ***ModelNet40***. Note that ULIP-2 has not released the checkpoint pre-trained by ShapeNet.
>
>
> #### Reference:
> #### [3] ShapeNet: An Information-rich 3D Model Repository. arxiv 2023.
> #### [4] ULIP: Learning a Unified Representation of Language, Images, and Point Clouds for 3D Understanding. CVPR 2023

---

> ### Author Response · Authors · 2023-11-22
> **Sincere Request for Further Discussions**
>
> Dear Reviewer Fjjw,
>
> Thanks again for your great efforts and constructive advice in reviewing this paper! With the discussion period drawing to a close, we expect your feedback and thoughts on our reply. We put a significant effort into our response, with several new experiments and discussions. We sincerely hope you can consider our reply in your assessment. We look forward to hearing from you, and we can further address unclear explanations and remaining concerns if any.
>
> Regards,
>
> Authors

---

> > ### Comment · Reviewer_Fjjw · 2023-11-22
> >
> > Thank you to the authors for their efforts. Here are my remaining concerns.
> > > We didn't claim the contrastive paradigm as a contribution in the paper.
> >
> > The claimed contribution such as the 3D multi-modal embedding space, is actually brought by the contrastive learning paradigm, which is similar to the learned one in ULIP. The main difference is that the authors align to the different model (ImageBind), which introduces additional modalities.
> >
> >
> > > Our motivation is not to develop a new separate model.
> >
> > I understand your points, but what I'm trying to say is that by giving a new name, Point-LLM, to the method of leveraging existing models (ImageBind-LLM), it may confuse readers into thinking that you have proposed a completely new model as your contribution. In fact, this approach is similar to learning a projection layer for utilizing ISS in 3D generalization, which you didn't actually rename. Therefore, when referring to Point-LLM, you could consider using the same strategy and modify your statement to indicate the incorporation of visual cache into ImageBind-LLM.
> >
> > > We show the ULIP-2 performance in the table below.
> >
> > I can't find the results of ULIP-2.
> >
> > > The teacher models of Point-Bind and ULIP are different in architectures and pre-training data
> >
> > Thank you for providing the new results. Firstly, Point-Bind should yield a result of 76.3 when using the same 3D backbone as Point-Bert. Secondly, it is evident that utilizing the same large image encoder greatly enhances performance. However, there still exists a significant difference in scale between the pre-training data, which might further boost performance.

---

> > > ### Author Response · Authors · 2023-11-22
> > > **Thanks for your reply!**
> > >
> > > Dear Reviewer Fjjw,
> > >
> > > Thank you for your response and additional constructive advice! We hope our further response can address your concerns.
> > >
> > > ---
> > > > **Q1: The claimed contribution (3D multi-modal embedding space) is actually brought by the contrastive learning paradigm**
> > >
> > > >
> > > Yes, we adopt this contrastive learning paradigm for pre-training, but present a 3D multi-modal embedding space ***for the first time***, which exhibits emergent capabilities for several 3D-centric multi-modal tasks. Our contributions mainly focus on such ***promising emergent capacities*** of the joint embedding space for 3D multi-modal learning, but ***not the pre-training technique*** to achieve it.
> > >
> > > Thanks for pointing out! As you suggested, in the revised manuscript, we have ***removed all the sentences*** that claim our joint embedding space as a contribution, and mainly highlight the significance of its emergent 3D multi-modal capabilities. We also clearly state that the adopted contrastive pre-training is based on ULIP's paradigm.
> > >
> > > ---
> > > > **Q2: Giving a new name, Point-LLM, may confuse readers into thinking that you have proposed a completely new model as your contribution**
> > >
> > > >
> > > Sorry for misunderstanding your point, and thanks for the valuable suggestion!
> > >
> > > As you suggested, when introducing Point-LLM, we have modified all the statements in the revised manuscript to ***always emphasize*** that our model is built upon a pre-trained ImageBind-LLM by incorporating it with Point-Bind, which is an emergent application without 3D instruction tuning, instead of a new separate model. This helps the readers to better understand our methodology and contributions.
> > >
> > > Furthermore, to explore the significance of Point-LLM for 3D domains, we have verified the potential of Point-LLM to conduct ***scene-level understanding***. The original Point-LLM is designed for object-level question answering, since the Point-Bind is only pre-trained by single-object datasets. We further implement a scene-level variant, and show the qualitative examples on ScanNet [1] in ***Figure 13 of Appendix***.
> > >
> > > Specifically, to obtain the scene-level understanding capacity, we fine-tune our object-level Point-LLM by an existing 3D question-answering dataset [2] constructed from ScanRefer [3]. We simply append three MLP layers with residual connections after Point-Bind’s 3D encoder, which are responsible for learning the scene-level 3D geometries. We only enable the new MLP layers to be trainable, while keeping other components frozen to preserve the pre-trained cross-modal knowledge.
> > >
> > > This experiment indicates that, although Point-LLM is extended from the existing ImageBind-LLM, our approach can be utilized as a base model to be fine-tuned for more complex 3D scenarios, demonstrating promising generalizability in 3D domains.
> > >
> > > ---
> > > > **Q3: I can't find the results of ULIP-2**
> > >
> > > >
> > > We are very sorry for this. The complete table of cross-modal retrieval is shown below. Note that ULIP-2 is pre-trained by a larger-scale dataset, Objaverse of 800K 3D shapes, which contributes to better 3D modality understanding. In contrast, our Point-Bind follows ULIP to pre-train on ShapeNet of only 53K samples. As shown, Point-Bind can outperform ULIP-2 on three cross-modal retrieval results, indicating the superior multi-modal alignment efficacy of our approach.
> > > | Method          | 3D → 3D | 2D → 3D | 3D → 2D | Text → 3D |
> > > |-----------------|---------|---------|---------|-----------|
> > > | PointCLIP       | 37.63   | 13.12   | 5.28    | 10.86     |
> > > | PointCLIP-V2    | 47.94   | 20.48   | 9.22    | 52.73     |
> > > | ULIP            | 60.58   | 20.30   | 29.75   | 50.51     |
> > > | ULIP-2 |**64.35** |19.21 | 31.63 | 57.05 |
> > > | Point-Bind  | 63.23 | **34.59** | **42.83** | **64.50** |
> > >
> > > ---
> > > > **Q4: Point-Bind should yield a result of 76.3 when using the same 3D backbone as Point-BERT**
> > >
> > > >
> > > Sorry for the typo, and thanks for pointing it out! The Point-Bind with a Point-BERT encoder should yield 76.3\% accuracy.
> > >
> > > ---
> > > > **Q5: There still exists a significant difference in scale between the pre-training data**
> > >
> > > >
> > > Thanks for the question. Note that, ***ImageBind freezes the ViT-H image encoder and text encoder of OpenCLIP during its pre-training***. That is, ImageBind and OpenCLIP share the same weights in their image and text encoders. Therefore, the ViT-H image encoder used to pre-train ULIP is ***exactly the same as*** that used to pre-train our Point-Bind.
> > >
> > > ####
> > > #### Reference:
> > > #### [1] ScanNet: Richly-annotated 3D Reconstructions of Indoor Scenes. CVPR 2017.
> > > #### [2] Chat-3D: Data-efficiently Tuning Large Language Model for Universal Dialogue of 3D Scenes. arxiv 2023.
> > > #### [3] Scanrefer: 3d object localization in rgb-d scans using natural language. ECCV 2020.
> > > ####
> > >
> > > Regards,
> > >
> > > Authors

---

> > > > ### Comment · Reviewer_Fjjw · 2023-11-22
> > > >
> > > > I would like to express my gratitude to the authors for their efforts in the rebuttal. As a result, I will increase my rating to 6.
> > > >
> > > > I encourage the authors to make the corresponding model and weights open source, including those mentioned in this rebuttal.

---

> > > > > ### Author Response · Authors · 2023-11-23
> > > > > **Thanks for your reply!**
> > > > >
> > > > > Dear Reviewer Fjjw,
> > > > >
> > > > > Thank you for acknowledging our response and efforts! We appreciate your valuable time and constructive comments.
> > > > >
> > > > > We will release the code and checkpoint as soon as possible. Thanks for your suggestion!
> > > > >
> > > > > Regards,
> > > > >
> > > > > Authors

---

### Official Review · Reviewer_gd23 · 2023-10-30

**Soundness:** 3 good
**Presentation:** 4 excellent
**Contribution:** 3 good
**Rating:** 8
**Confidence:** 5

**Summary:**

In their paper, the authors introduce two novel frameworks, Point-Bind and Point-LLM, for achieving 3D multi-modality understanding and generation. Point-Bind is a pretraining framework that unifies the semantic space across image, text, audio, and 3D data. Building on this foundation, the authors connect Point-Bind with the previously proposed ImageBind-LLM to incorporate 3D features into the LLM. This integration enhances the model's ability to understand and reason about multi-modal scenes.

**Strengths:**

The paper has several notable strengths:
1. The paper is well-written and easy to understand. All the approaches are based on the ImageBind.
2. Experiments are clear to support the claims of contributions and motivations.
3. The Point-Bind collects a new 3D-image-text-audio dataset and gets the best performance in 3D open-world Understanding.
4. The fine-tuning of Point-LLM is data-free, which requires no 3D instruction data.

**Weaknesses:**

1. While the paper's experimental design is well-executed, it is true that there are other 3D representation methods that could be explored to further validate the robustness and transferability of the proposed methods. For example, the authors could consider using JM3D [1] and CG3D [2] as additional 3D representation methods to be incorporated into their proposed frameworks. This would provide a more comprehensive evaluation of the proposed methods and help to establish their generalizability across different 3D representation methods.

2. The paper claims to introduce a “new 3D-image-text-audio dataset”, yet it appears to be a fusion of the existing ShapeNet55 and ESC-50 datasets.

3. One of the insights presented in the paper is the arithmetic of multi-modalities. Conducting additional quantitative experiments could further elucidate and solidify this contribution, thereby substantiating this claim.

4. The paper could be enhanced by expanding on the related work section, including references [3, 4] which have been missed. Including these references can provide a broader context and better situate the presented work within the existing body of knowledge.

[1] Beyond First Impressions: Integrating Joint Multi-modal Cues for Comprehensive 3D Representation, ACM MM23

[2] Clip goes 3d: Leveraging prompt tuning for language grounded 3d recognition, ICCV 23

[3] JM3D & JM3D-LLM: Elevating 3D Representation with Joint Multi-modal Cues, arXiv:2310.09503.

[4] 3D-LLM: Injecting the 3d world into large language models[J]. arXiv preprint arXiv:2307.12981.

**Questions:**

1. As the paper mentions, the final training data comprises only 9 categories from the public portions of ShapeNet55 and ESC-50, which seems insufficient for a pretrained work. Could this limitation impact the robustness of the proposed model?

2. The 3D question-answering capability of Point-LLM appears more akin to narrative construction rather than straightforward answering. Could you provide examples of scene understanding, such as outdoor scenes from autonomous driving contexts or indoor scenes from S3DIS, to substantiate the model's 3D question-answering ability?

3. The extent to which Point-Bind can benefit downstream tasks remains unclear. Providing more qualitative results, as opposed to solely textual descriptions, could offer a better understanding of its efficacy.

---

> ### Author Response · Authors · 2023-11-21
> **Response to Reviewer gd23 (Part #1)**
>
> We sincerely appreciate your valuable and careful reviews. We hope our response can address your concerns.
>
> ---
> > **Q1: The authors could incorporate JM3D and CG3D into the framework for a more comprehensive evaluation**
>
> >
> Thanks for your advice! It would be very helpful to evaluate the generalizability of our approach by incorporating techniques from other advanced methods like JM3D and CG3D.
> 1. **JM3D** proposes two delicate approaches to enhance the multi-modal pre-training of 3D models: Structured Multimodal Organizer (SMO) and Joint Multi-modal Alignment (JMA). SMC adopts multi-view rendered images and hierarchical text for more comprehensive representation, and JMA aims to achieve better mult-modal synergy by generating joint vision-language features. We also add the two techniques in JM3D into our Point-Bind for the image and text modalities within ImageBind, and evaluate on two benchmarks: 3D zero-shot classification and cross-modal retrieval on ModelNet40. As shown in the table below, the capabilities of Point-Bind are well enhanced by integrating SMO and JMA, indicating the importance of more comprehensive vision-language guidance.
>
> | Method         | Zero-shot Cls.| 3D → 3D  | 2D → 3D  | 3D → 2D  | Text → 3D |
> | -------------- | ---- | -------- | -------- | -------- | --------- |
> | Point-Bind | 78.0 | 63.2 | 34.6 | 42.8 | 64.5 |
> | Point-Bind w JM3D | **78.4** | **64.1** | **35.5** | **43.9** | **64.7**|
>
>
> 2. **CG3D** shares a similar contrastive learning paradigm with ULIP, and introduces learnable visual prompts for CLIP's image encoder for better adaption of 2D rendered images. For our Point-Bind, we also add learnable visual prompts to the image encoder of ImageBind, and report the results in the table below. On both benchmarks, the prompting approach from CG3D can improve the performance of Point-Bind, which demonstrates the effectiveness of fine-tuning the pre-trained image embeddings.
>
> | Method         | Zero-shot Cls.| 3D → 3D  | 2D → 3D  | 3D → 2D  | Text → 3D |
> | -------------- | ---- | -------- | -------- | -------- | --------- |
> | Point-Bind | 78.0 | 63.2 | **34.6** | 42.8 | 64.5 |
> | Point-Bind w CG3D | **78.2** | **63.5** | 34.3 | **43.2** | **64.8**|
>
> ---
> > **Q2: The claimed “new 3D-image-text-audio dataset” appears to be a fusion of ShapeNet55 and ESC-50**
>
> >
> Thanks for pointing out! Yes, our multi-modal pre-training dataset is constructed by combining existing datasets. We have rectified our claim in the revised manuscript by removing the word "new".
>
> ---
> > **Q3: Conducting quantitative experiments for "arithmetic of multi-modalities" could further solidify this contribution**
>
> >
> Thanks for your advice! It is quite necessary to show quantitative results to evaluate the multi-modal arithmetic ability of Point-Bind. As there is still no existing benchmark for 3D multi-modal arithmetic, we construct a new multi-modal arithmetic retrieval benchmark, which contains 200 multi-modal data pairs selected from ModelNet40, ESC-50, and ImageNet datasets.
> The data pairs are composed of three types, i.e., 3D-Text-Image, 3D-Audio-Image, 3D-Image-Image, for which we utilize the addition of the former two modalities to retrieve the last one. The retrieval patterns must make scene in real life, e.g., Bottle (3D) + “Car” (Text) -> Bottles in the car (Image), Tent (3D) + Water (Audio) -> Tents beside the river (Image), Person (3D) + Motor (Image) -> Human on the motorbike (Image).
>
> We evaluate Point-Bind on our proposed benchmark, and obtain the retrieved results by ranking feature similarities. As reported in the table below, pre-training with more modalities achieves better performance, which enhances the superior multi-modal alignment and cross-modal understanding capacity.
> |Text                 | 3D   | Image | Audio |    Acc. |
> | ---- | ----- | ----- | -------- | -------------- |
> | ✓  | ✓ | -    | -     | 44.5 |
> | ✓  | ✓ | ✓ | -    | 53.1% |
> | ✓  | ✓ | ✓ | ✓ | 56.4% |

---

> ### Author Response · Authors · 2023-11-21
> **Response to Reviewer gd23 (Part #2)**
>
> ---
> > **Q4: It's better to include more references [1, 2] in the related work section**
>
> >
> Thanks for pointing out! In the original submission, we have cited 3D-LLM [1] in the second paragraph of the related work section. In the revised manuscript, we have also added the discussion of JM3D & JM3D-LLM [2] in the same paragraph. Note that both papers are concurrent to our work.
>
> 1. **3D-LLM** injects 3D worlds into LLMs by rendering 3D scenes into multi-view images and construct 3D features by lifting the encoded 2D features. It targets on scene-level 3D understanding and generates a high-quality 3D-language dataset for tuning. Different from 3D-LLM, our Point-LLM directly encodes 3D object-level features in 3D space, and do not need the process of 3D instruction tuning. Also, Point-LLM can process 3D-centric multi-modal instructions for cross-modal reasoning.
>
> 2. **JM3D & JM3D-LLM** aligns 3D semantics with LLMs by object-level description data sourced from Cap3D [3]. Based on the pre-trained representations of JM3D, JM3D-LLM exhibits superior performance for detailed 3D captioning. Our Point-LLM is also different from it, since we do not need specific 3D instruction tuning and can conduct 3D multi-modal question answering emergent from Point-Bind.
>
> ---
> > **Q5: The final training data comprises only 9 categories from the public datasets**
>
> >
> Sorry for the confusion caused. We utilize the full 55 categories from ShapeNet for pre-training, in which only 9 categories contain the paired audio data from ESC-50 [4]. For the other 45 categories, we only calculate the contrastive loss guide by the image and text encoders of ImageBind, without using the audio modality. We have revised the manuscript to make this point more clear.
>
> ---
> > **Q6: Point-LLM appears more akin to narrative construction rather than straightforward answering**
>
> >
> Thanks for pointing out! As shown in ***Figure 12 of Appendix***, we show more examples of Point-LLM for straightforward question answering, e.g., "How to start it?”, “What is the purpose of this thing?". Our model can respond with precise answers that correspond to the input point cloud.
>
> #### Reference:
> #### [1] 3D-LLM: Injecting the 3D World into Large Language Models. arxiv 2023.
> #### [2] JM3D & JM3D-LLM: Elevating 3D Representation with Joint Multi-modal Cues. arxiv 2023.
> #### [3] Scalable 3D Captioning with Pretrained Models. arxiv 2023.
> #### [4] ESC: Dataset for Environmental Sound Classification. ACM MM 2015.

---

> ### Author Response · Authors · 2023-11-21
> **Response to Reviewer gd23 (Part #3)**
>
> ---
> > **Q7: Could you provide examples of scene understanding, such as outdoor or indoor scenes?**
>
> >
> Thanks for your advice! Our Point-LLM in the original submission is mainly designed for object-level question answering, since the Point-Bind is only pre-trained by single-object datasets. As you suggested, we further implement a scene-level variant of our model, termed Point-LLM$_{Scene}$. Due to the limited time period of the rebuttal, we focus on the understanding of indoor scenes on ScanNet [5], and show the qualitative examples in ***Figure 13 of Appendix***.
>
> Specifically, to obtain the scene-level understanding capacity, we fine-tune our object-level Point-LLM by an existing 3D question-answering dataset [9] constructed from ScanRefer [10]. We add three MLP layers with residual connections between Point-Bind’s 3D encoder and the LLM, which is responsible for learning the scene-level 3D geometries. We only enable the new MLP layers to be trainable, while keeping other components frozen to preserve the pre-trained cross-modal knowledge.
>
>
>
> ---
> > **Q8: The extent to which Point-Bind can benefit downstream tasks remains unclear**
>
> >
> Thanks for your advice! To evaluate our efficacy on downstream tasks, we regard Point-Bind as a pre-trained model, and fine-tune it for 3D shape classification without voting on ModelNet40 and ScanObjectNN datasets. As shown in the table below, compared to the original I2P-MAE and Point-BERT, the multi-modal learning of Point-Bind can effectively enhance their classification accuracy. This indicates the promising generalization ability of Point-Bind for assisting downstream 3D applications.
> | Method       | ModelNet40 | ScanObjectNN |
> | -------------- | ---- | ------- |
> | Point-BERT  | 92.7 | 83.1 |
> | Point-Bind | 93.4 |84.8 |
>
> | Method       | ModelNet40 | ScanObjectNN |
> | -------------- | ---- | ------- |
> | I2P-MAE | 93.7 | 90.1 |
> | Point-Bind | 93.9   | 91.2   |
>
> #### Reference:
> #### [5] ScanNet: Richly-annotated 3D Reconstructions of Indoor Scenes. CVPR 2017.
> #### [6] Scanrefer: 3d object localization in rgb-d scans using natural language. ECCV 2020.
> #### [7] 3d shapenets: A deep representation for volumetric shapes. CVPR 2015.
> #### [8] Revisiting point cloud classification: A new benchmark dataset and classification model on real-world data. ICCV 2019.
> #### [9] Chat-3D: Data-efficiently Tuning Large Language Model for Universal Dialogue of 3D Scenes. arxiv 2023.
> #### [10] Scanrefer: 3d object localization in rgb-d scans using natural language. ECCV 2020.

---

> > ### Comment · Reviewer_gd23 · 2023-11-22
> > **Increase initial rating to 8**
> >
> > I appreciate the author's comprehensive responses and the effective resolution of my concerns. Recognizing its potential impact in the 3D representation field and in agreement with other reviewers' positive feedback, I suggest further refinements as per collective advice. I am increasing my rating to 8 and will continue to monitor any additional questions from other reviewers

---

> > > ### Author Response · Authors · 2023-11-22
> > > **Thanks for you reply!**
> > >
> > > Dear Reviewer gd23,
> > >
> > > Thank you for acknowledging our response and efforts! We're glad to hear that you're satisfied.
> > >
> > > Regards,
> > >
> > > Authors

---

> ### Author Response · Authors · 2023-11-22
> **Sincere Request for Further Discussions**
>
> Dear Reviewer gd23,
>
> Thanks again for your great efforts and constructive advice in reviewing this paper! With the discussion period drawing to a close, we expect your feedback and thoughts on our reply. We put a significant effort into our response, with several new experiments and discussions. We sincerely hope you can consider our reply in your assessment. We look forward to hearing from you, and we can further address unclear explanations and remaining concerns if any.
>
> Regards,
>
> Authors

---

### Official Review · Reviewer_fqYr · 2023-10-31

**Soundness:** 3 good
**Presentation:** 3 good
**Contribution:** 1 poor
**Rating:** 5
**Confidence:** 5

**Summary:**

This paper presents Point-Bind and Point-LLM, enhancing 3D multi-modality ability. Specifically, the Point-Bind adopts a contrastive learning paradigm to align the point cloud representation to other modalities' representations (the output of the pre-trained Image-Bind model). The Point-LLM further injects the 3D representation output by the Point-Bind to Large Language Model, facilitating many interesting downstream applications like the question answering with the 3D model.

**Strengths:**

1: The Point-Bind utilizes the strong ability of the multi-modal foundation model Image-Bind, empowering the 3D open-world understanding ability with other modalities.

2: This Point-Bind and Point-LLM can be used in many downstream tasks, e.g., 3D question and answering, any-to-3D generation, and 3D zero-shot understanding.

3: The vivid figures and visualizations convey the usability and flexibility of the 3D multi-modal models Point-Bind and Point-LLM.

**Weaknesses:**

The academic contributions are somewhat limited.

1: For "Aligning 3D with ImageBind", "Any-to-3D Generation" and "3D Open-World Understanding", aligning the 3D point feature to pre-trained foundation models developed for other modalities using the contrastive paradigm is a well-used technique, like the ULIP [1], CLIP^2 [2] and OpenShape [3]. This paper follows a similar pipeline to train the Point-Bind model. The major difference is the choice of foundation models, i.e., ImageBind or CLIP. Similarly, the other two mentioned contributions, "Any-to-3D Generation" and "3D Open-World Understanding," are also very natural downstream applications of the aforementioned technology. The "Any" ability is mainly powered by the Image-Bind model rather than the proposed method. Therefore, these parts don't make much contribution to me.

2: For "3D Embedding-space Arithmetic", developing the bridge between modalities for modality-specific feature injection and facilitating the LLM for other modalities are also well developed by the previous works, like the BLIP-2 [4] and MiniGPT4 [5]. The BLIP-2 uses pre-trained multi-modal encoders and only trains a transformer block to adapt the image to the LLM, and the MiniGPT4 further unleashes the interaction ability, which share similarities with the proposed Point-LLM. The major difference is also the point cloud modality.

Overall, I think this paper has technique contributions and could be used widely in the community. However, the paper uses many well-validated techniques, and the main difference is to further verify the conclusion on the point cloud modality. Therefore, I tend to reject this paper at this time.

[1] Xue, Le, et al. "ULIP: Learning a unified representation of language, images, and point clouds for 3D understanding." CVPR. 2023.

[2] Zeng, Yihan, et al. "CLIP2: Contrastive Language-Image-Point Pretraining from Real-World Point Cloud Data." CVPR. 2023.

[3] Liu, Minghua, et al. "OpenShape: Scaling Up 3D Shape Representation Towards Open-World Understanding." NeurIPS 2023.

[4] Zhu, Deyao, et al. "Minigpt-4: Enhancing vision-language understanding with advanced large language models." arXiv preprint arXiv:2304.10592 (2023).

[5] Li, Junnan, et al. "BLIP-2: Bootstrapping Language-Image Pre-training with Frozen Image Encoders and Large Language Models." ICML 2023.

**Questions:**

Please see the weakness part.

---

> ### Author Response · Authors · 2023-11-21
> **Response to Reviewer fqYr (Part #1)**
>
> We sincerely appreciate your detailed and insightful reviews. We hope our response can address your concerns.
>
> ---
> > **Q1: Aligning 3D with pre-trained models using a contrastive paradigm has been a well-used technique, e.g., ULIP**
>
> >
> Sorry for the confusion caused. ***We didn't claim the contrastive paradigm as a contribution in the paper.*** Indeed, we are inspired by previous works like ULIP to adopt the contrastive paradigm for pre-training, but the main contributions of Point-Bind are in two aspects as follows.
>
> 1. **A 3D multi-modal embedding space.** For the first time, we align 3D with more than two modalities into a joint embedding space. Previous works only verify that 3D can be connected with 2D images and language from CLIP, while our Point-Bind indicates the potential for more modalities including ***audio***, ***video***, ***depth***, and ***infrared data***. As shown in ***Figure 8 of Appendix***, we add the visualization of cross-modal retrieval between 3D and three new modalities (video, depth, and infrared data). This unified space is expected to expand the scope of 3D models to wider cross-modal scenarios.
>
> 2. **Emergent new applications without specific training.** Benefiting from the joint space, our Point-Bind naturally motivates several new cross-modal applications, which ***does not need domain-specific training***. For example, previously, to achieve question answering (QA) with 3D objects, one has to collect large-scale 3D data with paired QA, and spend many computation resources for training. Instead, with our Point-Bind, the 3D QA capability is totally emergent from the joint embedding space given an off-the-shelf 2D QA model, avoiding the expensive data collection and training (referring to our Point-LLM ***in the following Q4***). Likewise, we do not need specific 3D-audio training to achieve audio-referred 3D zero-shot classification, or specific 3D-video training for video-to-3D retrieval. Such an emergent ability of Point-Bind significantly lowers the bar for efficiently achieving many new cross-modal applications.
>
> Therefore, Point-Bind as the first work develops a joint embedding space for 3D multi-modality, and enables many emergent applications without domain-specific training. Besides, we also construct a 3D-image-audio-text paired dataset and introduce a triplet contrastive loss for pre-training, which can be viewed as a miner contribution.
>
>
> ---
> > **Q2: "Any-to-3D Generation" is mainly powered by ImageBind rather than the proposed method**
>
> >
> Thanks for pointing out. Within the joint 3D embedding space, the basic text/image/audio-to-mesh generation is mainly powered by encoders of ImageBind. However, by the 3D encoder of Point-Bind, we can achieve point-to-mesh generation, and further enable ***3D editing with multi-modal instructions***, as visualized in ***Figure 10 of Appendix***.
>
> For example, given a **3D airplane**, we can provide a **language instruction**, "Color the 3D shape in red", or a pure yellow picture as the **visual instruction**. Then, we respectively feed them into Point-Bind's 3D encoder and ImageBind's text or image encoder. Due to the joint embedding space, the generative decoder can incorporate their semantics and output the **airplane in red/yellow**. Likewise, given an ordinary 3D bench, we can provide instructions like "Modify the material to wooden". The model can correspondingly generate a wooden chair.
>
> Therefore, our Point-Bind plays an important role in "Any-to-3D Generation" by encoding input point clouds into the joint multi-modal embedding space.
>
>
>
> ---
> > **Q3: "3D Open-World Understanding" is a very natural downstream application**
>
> >
> As analyzed above in Q1, such ***naturally emergent applicability*** is exactly one of the contributions of Point-Bind, which alleviates the need for additional task-specific training. For 3D open-world understanding, Point-Bind not only achieves the best performance for traditional text-referred classification, but also explores a new task, audio-referred open-world understanding, which poses a good baseline for future 3D open-world learning.

---

> ### Author Response · Authors · 2023-11-21
> **Response to Reviewer fqYr (Part #2)**
>
> ---
> > **Q4: The major difference between Point-LLM and existing BLIP-2/MiniGPT-4 is the point cloud modality**
>
> >
> Sorry for the confusion caused. We have updated this part to be more clear in the revised manuscript. ***The methodology of Point-LLM is Very Distinct to BLIP-2 [1] or MiniGPT-4 [2]***. Our Point-LLM doesn't simply inject other modalities into LLMs, but proposes a more efficient paradigm by the joint embedding space of Point-Bind.  Besides the different 2D and 3D modalities, our novelty compared to existing BLIP-2 or MiniGPT-4 is three-fold as follows.
>
> 1. **Solve 3D-language alignment by converting it into easier 2D-language alignment.** Considering the paucity of accessible 3D-caption data, it's hard to directly align the 3D modality with LLMs like existing 2D methods using extensive image-caption data. Fortunately, our ***Point-Bind presents a joint embedding space between 3D and multi-modality.*** This allows us to still align LLMs with 2D using ImageBind's image encoder by sufficient image-caption data, and then replace the 2D encoder with Point-Bind's 3D encoder. A visual cache model is specifically adopted to alleviate the 2D-3D domain gap. By regarding 2D as an intermediary, our cross-modal alternative solution can efficiently bridge the gap between 3D and LLMs, while overcoming the paucity issue of 3D-caption data, which fully demonstrates the ***emergent advantage of Point-Bind***.
>
> 2. **Do not need 3D instruction data collection and tuning.** Existing 2D visual LLMs normally conduct a specific instruction-tuning process after the 2D-LLM alignment, which requires to collect high-quality 2D QA pairs. For example, LLaVA [3] collects images with object-level annotations from MSCOCO [4] and generates an instruction dataset using GPT-4. Such data collection and tuning processes are expensive and time-consuming, especially in the complex 3D domain. Fortunately, ***also by the joint embedding space of Point-Bind***, we still conduct the instruction tuning of Point-LLM in 2D space using the already collected 2D QA data. This contributes to superior data and computation efficiency.
>
> 3. **Process 3D-centric multi-modality instructions within LLMs.** Emergent from Point-Bind, our Point-LLM can accept instructions of both 3D and other multi-modality input, as shown in ***Figure 3 of the paper***. BLIP-2 and MiniGPT-4 can only tune LLMs to understand 2D images, but ***our joint embedding space*** enables Point-LLM to directly understand multi-modality instructions and reason their relationship without specific multi-modality QA training, e.g., responding to a question based on a point cloud and a picture or an audio.
>
> Therefore, instead of directly transferring the training pipeline of 2D visual LLMs into 3D, our Point-LLM proposes an alternative solution emergent from Point-Bind’s joint embedding space, which is both efficient and effective for 3D domains.
>
> #### Reference:
> #### [1] BLIP-2: Bootstrapping Language-Image Pre-training with Frozen Image Encoders and Large Language Models. arxiv 2023.
> #### [2] Minigpt-4: Enhancing vision-language understanding with advanced large language models. arxiv 2023.
> #### [3] Visual Instruction Tuning. NeurIPS 2023.
> #### [4] Microsoft COCO: Common objects in context. ECCV 2014.

---

> ### Author Response · Authors · 2023-11-22
> **Sincere Request for Further Discussions**
>
> Dear Reviewer fqYr,
>
> Thanks again for your great efforts and constructive advice in reviewing this paper! With the discussion period drawing to a close, we expect your feedback and thoughts on our reply. We put a significant effort into our response, with several new experiments and discussions. We sincerely hope you can consider our reply in your assessment. We look forward to hearing from you, and we can further address unclear explanations and remaining concerns if any.
>
> Regards,
>
> Authors